

# Forcing and Responses of the Surface Energy Budget at Summit, Greenland

Nathaniel B. Miller[1], Matthew D. Shupe[1], Christopher J. Cox[1], David Noone[2], P. Ola G. Persson[1], and Konrad Steffen[3]

[1]Cooperative Institute for Research in Environmental Science, University of Colorado and NOAA-ESRL, Boulder, Colorado.
[2]College of Earth, Ocean, and Atmospheric Sciences, Oregon State University, Corvallis, OR
[3]Swiss Federal Research Institute WSL, Birmensdorf, Switzerland.

*Correspondence to:* Nathaniel Miller (millernb@colorado.edu)

**Abstract.**

Greenland ice sheet surface temperatures are controlled by an exchange of energy at the surface, which includes radiative, turbulent and ground heat fluxes. Data collected by multiple projects are leveraged to calculate estimates of all surface energy budget (SEB) terms at Summit, Greenland for the full annual cycle from July 2013 - June 2014 and extend to longer periods for

the radiative and turbulent SEB terms. Radiative fluxes are measured directly by a suite of broadband radiometers. Turbulent sensible heat flux is primarily estimated via the bulk aerodynamic method, and the turbulent latent heat flux is calculated via a two level approach using measurements at 10 and 2 m. The subsurface heat flux is calculated using a string of thermistors buried in the snow pack. Extensive quality-control data processing produced a data set in which all terms of the SEB are present 75% of the full annual cycle, despite the harsh conditions. By including a storage term for a near surface layer, the SEB is balanced

in this data set to within the aggregated uncertainties for the individual terms. November and August case studies illustrate that surface radiative forcing is driven by synoptically forced cloud characteristics, especially by low-level, liquid-bearing clouds. The annual cycle and seasonal diurnal cycles of all SEB components indicate that the non-radiative terms are anti-correlated to changes in the total radiative flux, and are hence responding to cloud radiative forcing. Generally, the non-radiative SEB terms and the upwelling longwave radiation component compensate for changes in downwelling radiation, although exact

partitioning of energy in the response terms varies with season and near-surface characteristics such as stability and moisture availability. Substantial surface warming from low-level clouds typically leads to a change from a very stable to a weakly stable near-surface regime with no solar radiation or from a weakly stable to neutral/unstable regime with solar radiation. Relationships between forcing terms and responding surface fluxes show that the upwelling longwave radiation produces 55-75% (40-50%) of the total response in the winter (summer) and the non-radiative terms compensate for the remaining change

in the combined downwelling longwave and net shortwave radiation. Because melt conditions are rarely reached at Summit, these relationships are documented for conditions of surface-temperature below 0°C, with and without solar radiation. This is the first time that forcing and response term relationships have been investigated in detail for the Greenland SEB. These results should both advance understanding of process relationships over the Greenland icecap and be useful for model validation.





# 1 Introduction

The exchange of energy at the Greenland Ice Sheet (GIS) surface must be thoroughly characterized to fully understand the processes that govern surface temperature variability, which is important in monitoring and modeling ice sheet mass balance (Box, 2013). Observations suggest near-surface temperatures are increasing; the GIS is trending toward greater spatial melt extent (McGrath et al., 2013) with increased melt runoff due to atmospheric warming (Hanna et al., 2008). The amalgamated freshwater runoff, in combination with ice discharge, determines how this major reservoir of Northern hemisphere ice affects freshwater input into the North Atlantic and Arctic Oceans, and subsequently, global ocean circulation and sea level rise. Surface melt processes currently account for approximately half of the total mass loss of the entire GIS (van den Broeke et al., 2009), and during prolonged periods of elevated surface temperatures this proportion is even greater (Smith et al., 2014). The melt process occurs in two steps. First, energy flux to the surface is used to increase the surface temperature. Then, after the melting point is reached, excess net surface energy flux is used to convert ice into liquid water. As an increasing area of the interior GIS approaches the melting point of snow in summer, spatial and temporal variations of the net surface energy flux are paramount in determining when the melting point is reached, over what spatial area this occurs, and the amount and rate of melt after this threshold is reached.

The surface energy budget (SEB) is a balance of radiative, turbulent and ground heat fluxes, which are coupled through a variety of processes. Once the surface temperature reaches the melting point of snow, additional energy goes toward melt, limiting the surface temperature to $0°C$. In the absence of phase change, however, a change in one of the SEB terms must be balanced by a change in another term or combination of terms. Importantly, the surface temperature is related to multiple SEB terms including upwelling longwave radiation, turbulent sensible heat, and ground heat fluxes. Over time scales long enough for the surface temperature to adjust, closure of the SEB is achieved and all of the energy exchange at the surface is accounted for. Because of the high emissivity (and hence high longwave absorptivity) of the snowpack, the surface is able to adjust relatively quickly to longwave influences (e.g., whether that is a warm cloud or a cold, clear sky). In contrast to its efficient ability to absorb longwave radiation, the GIS has a high shortwave albedo and reflects much of the incoming solar radiation. Liquid-bearing clouds are frequent above the GIS during summer (Shupe et al., 2013b) and have strong implications for increasing melt extent (Bennartz et al., 2013) and meltwater runoff (Van Tricht et al., 2016). In fact, clouds act to radiatively warm the central GIS throughout the year (Miller et al., 2015; Van Tricht et al., 2016), more than would occur via solar radiation acting alone, as a result of the year-round high surface albedo. Thus, the primary radiative influences on raising surface temperatures in this region are the solar zenith angle and occurrence of clouds.

A change in the downwelling radiative flux caused by clouds and/or solar radiation will induce a response of the atmospheric boundary layer and surface. Boundary layer depth and stability are influential for exchange processes, such as sublimation fluxes, which modulate accumulation (Berkelhammer et al., 2016). Miller et al. (2013) shows a degradation of the surface-based temperature inversion in the presence of liquid-bearing clouds, which impacts the near-surface stability (Hudson and Brandt, 2005) and thus turbulent mixing. A regional modeling case study by Solomon et al. (2016) indicates also that the response of turbulent and conductive heat fluxes to cloud radiative forcing is important when considering surface-atmosphere





interactions. Investigating these responses and interactions throughout the year is paramount in discerning the net effect of liquid-bearing clouds on surface temperatures and, consequently, on sub-surface temperatures and melt processes.

The central GIS is a massive reservoir of snow and ice, responding to energy changes at the surface by conducting heat into or out of the subsurface. Thus, the ice sheet damps the effects of either strong radiative warming or cooling at the atmosphere-snow

interface. Warmer subsurface temperatures, resulting from warming of surface temperatures, can change the snow morphology and precondition the surface to have less capacity for removing subsequent heat excesses generated by atmospheric processes (Solomon et al., 2016). Proper atmosphere/ice sheet coupling is important to allow for physically realistic radiational cooling at the surface, in order to minimize surface temperature biases in forecast models (Dutra et al., 2015).

Regional and global climate models are a critical tool for understanding the fate of the GIS and attempt to capture the

non-trivial interactions between the atmosphere and the GIS. Early studies parameterized the SEB of the GIS using meteorological measurements from summer camps in western Greenland and observations of albedo from satellites (van de Wal and Oerlemans, 1994; Konzelmann et al., 1994). More recently, computationally advanced, fully coupled, climate models project enhanced surface melt as GIS surface temperatures increase under future $CO_2$ forcing scenarios (Vizcaíno et al., 2014). Yet, these state-of-the-art climate models have surface temperature biases over the GIS, likely due to the under representation

of liquid-bearing clouds (Kay et al., 2016). To better understand and represent the important processes that currently hinder models, detailed surface-based observations are valuable.

In western Greenland, detailed measurements of the surface mass balance (van de Wal et al., 2005), surface radiation balance (van den Broeke et al., 2008), and surface energy balance (van den Broeke et al., 2011) have been reported, all of which focus on the ablation zone. In central Greenland, the most sophisticated and comprehensive long-term observations of surface energy

budget are made at Summit Station. While a majority of the published literature has focused on the summer season (Cullen and Steffen, 2001; Kuipers Munneke et al., 2009) some studies have targeted SEB annual cycles in 2000-2001 (Cullen, 2003) and 2001-2002 (Hoch, 2005). In addition, various studies have focused on specific components of the SEB, such as surface latent (Box and Steffen, 2001) or sensible heat (Cohen et al., 2007; Cullen et al., 2007; Drüe and Heinemann, 2007) fluxes. Annual surface radiation fluxes have been reported at Summit by van den Broeke et al. (2008), Cox et al. (2014), and Miller

et al. (2015), as well as longwave flux divergence in the boundary layer by Drüe and Heinemann (2007) and Hoch et al. (2007). Yet, prior to May 2010 there have been limited ground-based measurements of the atmospheric state and cloud properties to complement these temporally sporadic SEB investigations and to support process-based understanding of SEB variability on time scales from minutes to seasons.

This study uses comprehensive ground-based measurements to investigate interactions between the atmosphere and the

central GIS throughout the year in order to understand how energy exchange drives temporal variability in surface temperature. Summit Station is currently within the accumulation zone, recording only 2 melt events since 1889 (Nghiem et al., 2012). The lack of melt events provides the opportunity to examine relationships between the various surface energy fluxes in all seasons without the energetic influence of phase change at the surface. Using a unique compliment of data/measurements at 30-minute temporal resolution, we present a pair of case studies to illustrate cloud effects on the balance of energy at the surface and,

consequently, the subsurface snow in central Greenland. Next, we characterize the annual and diurnal cycles of the radiative,





turbulent and conductive heat fluxes for one year and evaluate SEB closure. Finally, we investigate the seasonal responses of the turbulent heat fluxes, subsurface heat flux, and upwelling longwave flux to changes in downwelling longwave and net shortwave fluxes, establishing process-based energy flux relationships.

## 2 Measurements and Methods

Near-surface instrumentation at Summit Station (72°N 38°W, 3211 m) is used to characterize the surface energy budget in central Greenland. Net radiative (Q), turbulent sensible (SH), turbulent latent (LH), and total subsurface (G) heat fluxes determine the net surface flux ($F_s$) according to the equation:

$$F_s = Q + SH + LH + G. \tag{1}$$

The total subsurface heat flux (G) considered here is a combination of the conductive heat flux (C) and heat storage in a near-
surface layer (S), detailed in Section 2.5. Each of these four terms is defined such that a positive value sends energy towards the surface and visa versa. For all measurements described here, a 30-minute time window is used; this time window was chosen to fit the constraints set by eddy covariance calculations for sensible turbulent flux (Section 2.3), but is sufficiently brief to capture both the diurnal cycle and the SEB response to atmospheric variability of interest here.

All SEB terms are estimated for 75.3% of an annual cycle, spanning July 2013 – June 2014, although Q, SH and LH are
also measured prior to July 2013. The techniques used to calculate each SEB term, the data availability periods, and associated uncertainties are outlined in the following sub-sections. The estimated uncertainty in each SEB term is summarized in Table 1. While each component of the SEB has it's own uncertainty, at times the various estimates use the same input and are thus not independent. For example the longwave measurements are used to derive the skin temperature, which is input into both the bulk sensible heat flux and conductive heat flux estimates.

**Table 1.** Estimated uncertainty in each surface energy budget term.

| LW↓ or LW↑ | SW↓ or SW↑* | SH | LH | C | S |
|---|---|---|---|---|---|
| 1.0 W m$^{-2}$ | 1.8% (> 1.0 W m$^{-2}$) | 8.7 W m$^{-2}$ | 60% | 26% | 80% |

* SW↑ in 2014 = 2.8% (> 1.0 W m$^{-2}$)

## 20 2.1 Measurements

Redundancy of many direct measurements used to derive the SEB components is imperative in the harsh Arctic environment where frost, rime and extreme cold create operational challenges. Certain measurement techniques are only valid during specific atmospheric conditions and operational temperature ranges of the instrumentation. As a result, redundant data streams and multiple independent methodologies are considered whenever possible to investigate suspected biases and fill in data gaps
during instrument downtime.



Twice daily radiosondes (0 and 12 UTC) from the Integrated Characterization of Energy, Clouds, Atmospheric State, and Precipitation at Summit (ICECAPS, Shupe et al., 2013b) project are used to directly measure the atmospheric temperature and humidity profile. A near-surface meteorological tower, maintained by the National Oceanic and Atmospheric Administration's Global Monitoring Division (NOAA/GMD), is the primary source of the near-surface ($\approx$ 2 m and $\approx$ 10 m) temperature

measurements with a specified resolution of 0.1°C. An experiment on Closing the Isotope Balance at Summit (CIBS), approximately 1 km northeast of the NOAA tower, included a broad suite of advanced meteorological measurements for evaluating surface exchange processes, including aspirated temperature measurements at 2 and 10 m. The CIBS instruments were mounted on a 50 m tower operated by the Swiss Federal Institute of Technology (ETH) Zürich. On average the CIBS 2 m temperatures are 0.72°C greater than the NOAA/GMD 2 m temperatures with a Root Mean Square (RMS) difference of 1.64°C. A portion of

the RMS difference is due to spatial distance between measurement locations and possibly also due to local variability in snow accumulation which would lead to differences in the measurement heights of the sensors. In addition, CIBS included Metek USA1 three-dimensional ultra sonic anemometers to directly measure orthogonal components of high frequency fluctuations in temperature and wind speed. The sonic anemometers (20 Hz sampling rate), equipped with heated transducers to prevent riming or frost buildup, were mounted at 2 and 10 m on the 50-m tower. Before 19 January 2013 the heaters operated only when

there were significant data dropouts due to rime/frost; after this date the heaters were on constantly. Comparison of the data before and after the heater configuration change indicate that sensible fluxes generated by the heating elements are sufficiently small that they are well within the measurement uncertainty. The high frequency sonic anemometer windspeed measurements are averaged to estimate the mean 30-minute wind speed. Redundant wind speed measurements are also made by CIBS cup anemometers, which have moving parts that have a frictional threshold that requires a wind speed of at least $0.5 \, \mathrm{m\,s^{-1}}$ for

reliable measurements. Comparisons between the two measurements for conditions above $0.5 \, \mathrm{m\,s^{-1}}$ show a RMS difference of $1.75 \, \mathrm{m\,s^{-1}}$ and a bias of $-0.55 \, \mathrm{m\,s^{-1}}$ in the cup anemometer data.

Subsurface temperatures are measured by Campbell Scientific 107 Temp Probes buried in the snow (every 20 cm in depth) near the 50-m tower. The height of the surface relative to the thermistor string is estimated from a downward facing sonic ranger mounted on the tower above the thermistor string. During the single year when the thermistor data was available (July

2013 – June 2014) the surface height increased by 0.68 m. Due to scatter in the reported surface heights, the snow depths are smoothed using a 5-day running window to remove erroneous spikes in the snow depth. Realistic longer term discontinuities due to actual snow events were maintained by limiting the period over which data smoothing occurred. Inexplicably, on 27 May 2014 the sonic ranger reported an abrupt 17.8 cm decrease in the surface height. The near surface thermistor variability indicates that this was unrealistic, hence an offset of -17.8 cm was applied to the thermistor depths thereafter through the end

of the study period. The standard deviation over 30-minutes of the 1-minute subsurface temperature data indicates that the variability decays as a function of depth because of a decline in the thermal effects of wind ventilation and direct solar heating due to solar penetration. To minimize the impact of these complicating issues a standard deviation threshold of 0.1 is used to determine that the acceptable minimum depth to use for the shallowest subsurface thermistor is about -20 cm.

The specific humidity at 2 and 10 m, which is needed for deriving LH, is calculated from CIBS relative humidity and temper-

ature measurements in combination with NOAA/GMD temperature and pressure measurements. From July 2012 to Dec 2013





direct measurements of water vapor mixing ratio are obtained via a Picarro model L2120 spectrometer, which was calibrated using a LiCor LI160 dew point generator (Bailey et al., 2015). The instrument directly samples air moisture content once an hour at multiple levels on the 50-m tower using a constrained inlet system to limit large (> 50 µm) hydrometers from being incorporated into the vapor measurements. Comparing meteorologically derived specific humidity values at approximately 1-2

m and 9-10 m above the surface to the highly accurate Picarro measurements reveals a small bias of +0.065 $\mathrm{g\,kg^{-1}}$. The percent error, using the Picarro measurements as truth, at the 2 and 10 m levels are 53% and 30%, respectively.

Investigating the surface flux estimates in combination with active and passive cloud property measurements yields a comprehensive understanding of how clouds affect the GIS energy budget. In addition to the aforementioned radiosondes, ICECAPS also measures the cloud properties via a comprehensive suite of instruments, in operation since May 2010. ICECAPS is de-

scribed in detail by Shupe et al. (2013b). Liquid water path (LWP) and precipitable water vapor (PWV) are estimated using a physical retrieval via a pair of microwave radiometers, similar to Turner et al. (2007). In a dry environment, such as Summit, it is advantageous to use a total of 3 channels (23.84, 31.40, 90.0 GHz) to increase sensitivity and effectively reduce uncertainty (LWP ≈ 5 $\mathrm{g\,m^{-2}}$, PWV ≈ 0.35 mm) (Crewell and Löhnert, 2003). The primary changes to the LWP values estimated in Miller et al. (2015) are an improved liquid-water model (TKC, Turner et al., 2016) and the use of three channels in the retrieval instead

of four. By excluding the 150.0 GHz channel, biases in LWP retrievals due to precipitating ice hydrometers will not impact the overall statistical results (Pettersen et al., 2016). Vertically resolved cloud presence is determined by a 35-GHz Millimeter Cloud Radar (MMCR).

## 2.2 Radiative Flux

Four broadband radiation components comprise the net radiation at the surface (Q):

$$Q = LW\!\downarrow - LW\!\uparrow + SW\!\downarrow - SW\!\uparrow. \tag{2}$$

At Summit Station ETH maintains broadband radiative flux measurements, at approximately 2 m above the surface. Kipp and Zonen CG4 pyrgeometers measure the upwelling and downwelling thermal emission (LW↑ and LW↓) in the spectral range of 4.5 − 40 $\mu$m and Kipp and Zonen CM22 pyranometers measure the upwelling and downwelling solar irradiance (SW↑ and SW↓) in the spectral range of 200 − 3600 nm. In this study the radiative flux measurements extend from January 2011 - June

25    2014.

Data processing for radiation measurements used here is similar to Miller et al. (2015), including corrections to the LW↓ components based on the net longwave radiation and comparison to colocated broadband radiation measurements operated by NOAA-GMD. An estimated Gaussian longwave radiation measurement uncertainty of 4-5 $\mathrm{W\,m^{-2}}$ (Gröbner et al., 2014) for the 1-minute data translates to about 1 $\mathrm{W\,m^{-2}}$ uncertainty in the 30-minute average data. Assuming an emissivity uncertainty

of 0.005 a LW-derived surface temperature has an approximate uncertainty of 0.6°C, which is derived from the radiation measurements thusly:

$$T_{surf} = [(LW\!\uparrow - (1 - \epsilon)\,LW\!\downarrow)/(\epsilon\sigma)]^{0.25}, \tag{3}$$



where surface emissivity($\epsilon$) = 0.985 and $\sigma =$ is the Stefan-Boltzmann constant. Comparing LW↑ to similar, proximate NOAA/GMD radiation measurements indicates that there is general agreement within the estimated 4-5 W m$^{-2}$ uncertainty of the longwave radiative components. Yet, for very cold surface temperatures (i.e., < -46°C) differences between the NOAA/GMD and ETH LW↑ are more pronounced. Hence, a third degree polynomial was used to fit the difference between the ETH and NOAA/GMD

LW↑ as a function of the ETH LW↑. A correction factor (y) was applied based on the measured ETH LW↑ (x) value according to the equation: $y = -14.99 + 0.1715x - 0.000668x^2 + 8.579e - 7x^3$, which assumes the more recently calibrated NOAA/GMD pyrgeometers are accurate. After applying the adjustments to LW↑ and LW↓ (Miller et al., 2015) the 1-minute LW data are consistent with a total uncertainty of 4-5 W m$^{-2}$.

The surface albedo (SW↑/SW↓) is affected by the solar zenith angle, and for clear-sky days should have a minimum at

solar noon. During 2014 an asymmetry in the diurnal cycle is observed in the measured albedo, where the albedo in the morning is up to 10% lower than in the evening. The NOAA/GMD measurements, which are mounted on the same fixed arm, indicate the same issue (possibly a gradual slope to the surface due to snow drifts). There is good agreement between the SW↓ measurements and the total direct plus diffuse SW↓ values when available, suggesting that this issue is unlikely a leveling problem in the SW↓ component. Hence, the SW↑ value is estimated in 2014 using the SW↓ value according to the equation:

$SW↑ = \alpha SW↓$, where $\alpha$ is the albedo. A linear relationship between albedo and solar zenith angle (Z) for 2011 − 2013 is used to estimate an albedo in 2014 according to the equation: $\alpha = 0.798 + 0.00107 * Z$. Comparing the measured SW↑ to the parameterized SW↑ yields an RMS difference of 5.7 W m$^{-2}$ for SW↓ < 278 W m$^{-2}$ and 12.6 W m$^{-2}$ for SW↓ > 278 W m$^{-2}$. Thus, the uncertainty in the parameterized SW↑ component is ≈ 5.7 W m$^{-2}$ for small sun angles and ≈ 2.8% for larger SW↓ values. These uncertainty estimates are larger than the reported uncertainty in the measured SW components of

1.8% (Vuilleumier et al., 2014).

During periods of 2013 and 2014 the SW↓ component has a bias that is evident when the sun is below the horizon, hypothesized to be due to a grounding issue. A bias correction of 2.45 W m$^{-2}$ was applied to 20 November 2013 to 30 January 2014, determined by the average value when the solar zenith angle was greater than 95°. From 31 January 2014 to 14 April 2014 a bias correction of 4.61 W m$^{-2}$ is applied to the SW↓ to remove the negative bias.

## 2.3   Turbulent Sensible Heat Flux

The net surface flux is influenced by the temperature of the overlying air, i.e. warmer near-surface air will increase the sensible heat transferred to the surface. Direct heat transfer, via conduction, from the atmosphere to the snowpack is only prominent very close to the surface, thus heat is primarily transferred via turbulent eddies. These eddies act to mix the air within the surface layer, reducing the vertical temperature gradient. Estimates of the sensible heat flux are calculated using two independent

methods: eddy correlation method and the bulk aerodynamic method.





The eddy correlation (EC) method (e.g., Oke, 1987) calculates the covariance between the anomalies in the vertical wind ($w'$) and temperature ($\theta'$) to determine the turbulent sensible heat flux according to the equation:

$$SH = \rho c_p \overline{w'\theta'}, \tag{4}$$

where the constants are the density ($\rho$) and heat capacity ($c_p$) of air. By using direct measurements of windspeed and tempera-
ture from a three-dimensional sonic anemometer, an accurate calculation of the heat exchange at $\approx 2$ m is obtained.

A 30-minute averaging period is a short enough time-window to exclude issues of non-stationarity while still long enough to include low frequency contributions to the turbulent heat flux. Various quality-control (QC) measures are implemented to ensure the data is representative of the entire sensible heat flux during the 30-minute window. QC measures exclude large changes in windspeed or wind direction, upwind contamination by the experimental apparatus, and $\pm$ 30% deviations from
characteristic -5/3 slope in the inertial subrange (Kaimal et. al. 1972). Applying the QC criteria flags 75% of the available data, spanning September 2011 – June 2014. Thus, for the 85% of this period that either have instrument downtime or where the data are QC flagged, an alternative approach is used.

Due to the limited data set available from the EC method, a bulk aerodynamic method is also used in order to fill in data gaps for the time period June 2011 – June 2014. The bulk transfer method uses Monin-Obukhov similarity theory to estimate
turbulent sensible heat flux at the surface:

$$SH = \rho c_p C_h U (T_{surf} - T_{2m}), \tag{5}$$

where $U$ is the mean horizontal wind speed at 2 m, $T_{surf}$ is the skin temperature, $T_{2m}$ is the temperature at 2 m, and $C_h$ is the sensible heat transfer coefficient for the 2 m reference height (Persson et al., 2002; Fairall et al., 1996). NOAA/GMD meteorological data is the primary source of the 2 m temperature measurements and data gaps are filled with CIBS temperature
data. Cup anemometer measurements fill in data gaps of the sonic anemometer-derived 2 m windspeed measurements. $C_h$ is based on the roughness of the surface and assumes scalar velocity and temperature roughness lengths with corrections to account for boundary layer stability. An optimal (as compared to the EC measurements) velocity roughness length of 3.8e-4 m (Kuipers Munneke et al., 2009) and a roughness length for temperature of 1e-4 m (Andreas et al., 2005) are assumed constant in time. Separate stability correction functions for stable or unstable boundary layer conditions are used to iteratively coverge
on the best estimate sensible heat flux (Persson et al., 2002).

Comparing the bulk sensible heat flux to the quality controlled EC data gives an indication of the uncertainty in the bulk method. Bulk data is deemed valid when the surface friction velocity ($u_* = [-\overline{u'w'}]^{0.5}$) value exceeds $0.03\,\mathrm{m\,s^{-1}}$. A correlation coefficient of 0.89 exists between the two techniques for the subset of data deemed valid for both techniques. The RMS difference between the two methods ($8.7\,\mathrm{W\,m^{-2}}$) is the net estimated uncertainty in the sensible heat flux. Compared to the
EC data the bulk method has a bias of $+7.0\,\mathrm{W\,m^{-2}}$. For instances where the bulk sensible heat flux magnitude is less than 10 $\mathrm{W\,m^{-2}}$ the bias and RMS difference decrease to $+3.5$ and $2.60\,\mathrm{W\,m^{-2}}$, respectively. This improvement suggests some of the differences could be due to inaccurate stability correction functions, uncertainty in the surface temperature derived from LW





measurements and snow emissivity assumptions, or roughness length values. Sensible heat flux discrepancies could also be due to measurement height differences between the EC and bulk methods. While the bulk method uses the measured surface skin temperature the EC values are measured at 2 m, which could differ from the sensible heat flux directly at the surface under very stable conditions. This suggests that the true SH uncertainty is smaller than estimated here. The covariance $u_*$ and bulk $u_*$ are

well correlated (0.84) with a RMS difference of 0.55 $\mathrm{m\,s^{-1}}$ and the bulk values are biased low (-0.026 $\mathrm{m\,s^{-1}}$). Changing the velocity roughness length to 4.5e-4 m, which is that determined for snow-covered multi-year sea ice (i.e., Persson et al., 2002) increases the RMS differences for the sensible heat flux by 1.4 $\mathrm{W\,m^{-2}}$, suggesting that variability in the roughness of the surface could contribute to error in the bulk parameterization. A majority of the 8.7 $\mathrm{W\,m^{-2}}$ uncertainty in the bulk estimates is likely due to uncertainties in the skin temperature as estimated from a constant surface emissivity. From June 2011 to June

2014 the bulk estimates are available for 78% of the time period. Thus, filling in EC data gaps with the bulk values vastly improves the temporal coverage of the sensible heat estimates.

## 2.4   Turbulent Latent Heat Flux and Stability

Turbulent eddies also affect the surface energy budget by transferring latent heat toward or away from the surface. Frequently the specific humidity increases with height above the surface, resulting in a transfer of latent energy toward the surface possibly

resulting in deposition. The bulk method used by Persson et al. (2002) assumes saturation conditions at the surface, which is not always a valid assumption for dry snow (Albert and McGilvary, 1992). In central Greenland the two-level profile method has been shown to be superior to the bulk method (Box and Steffen, 2001) as it can account for sublimation and deposition to the surface.

The profile method used here is similar to Steffen and DeMaria (1996) such that the latent heat flux is calculated from

near-surface horizontal wind (U) and mixing ratio (q) gradients ($\Delta$ = value at 10 m - value at 2 m) according to the equation:

$$LH = \rho L_v k^2 z_r^2 \Big(\frac{\Delta U}{\Delta z}\frac{\Delta q}{\Delta z}\Big)(\phi_m\phi_e)^{-1}, \tag{6}$$

where $\rho$ is the density of air, $L_v$ is the latent heat of vaporization, $k$ is the von Kármán constant (0.4), and $z_r$ is the log mean height ($\frac{\Delta z}{\ln(z_2 z_1^{-1})}$). The stability functions for the transfer of momentum ($\phi_m$) and water vapor ($\phi_e$) are corrections based upon the stability of the boundary layer and will either increase (unstable conditions) or decrease (stable conditions) the surface flux.

A measure of boundary layer stability is attained via calculation of the bulk Richardson number (Ri). The sign of Ri indicates whether mechanical mixing (positive) or buoyancy (negative) is more important in producing turbulence. Ri is dependent on the gradient in virtual potential temperature ($\Delta\theta_v$), wind speed ($\Delta u$) and respective measurement heights ($\Delta z$) according to the equation:

$$Ri = \frac{g}{\bar{\theta_v}}\frac{\Delta\theta_v\Delta z^{-1}}{(\Delta u\Delta z^{-1})^2}, \tag{7}$$

where g is the acceleration due to gravity (9.81 $\mathrm{m\,s^{-2}}$ ) and $\bar{\theta_v}$ is the average virtual temperature (K) between the two levels. In accordance with Steffen and DeMaria (1996), Ri is used to calculate the stability corrections. Coefficients for relating Ri to the stability factors are obtained from a study conducted in 2000, which used eddy correlation turbulence measurements to



**Table 2.** Stability functions for unstable and stable regimes from Cullen (2003).

| Stability function | Unstable (Ri < 0) | Stable (0 < Ri < 0.25) |
|:---:|:---:|:---:|
| $\phi_m$ | $(1 + 27|Ri|)^{-0.2}$ | $(1 + \frac{4Ri}{1-4Ri})$ |
| $\phi_e$ | $(1 + 19|Ri|)^{-0.55}$ | $(1 + \frac{3Ri}{1-4Ri})$ |

obtain the relationships in Table 2 (Cullen, 2003). For stable Ri values greater than zero the stability functions act to reduce the magnitude of the latent heat flux. For Ri greater than the critical Richardson number (Ric = 0.25) vertical turbulence becomes small and, in theory, results in laminar flow. Grachev et al. (2013) indicates that intermittent and non-stationary turbulence can exist even in this super critical regime. Assuming LH = 0 for Ri > 0.25 could underestimate latent heat flux from intermittent

non-stationary turbulence but isotopic closure calculations indicate that for very-stable boundary layers tracers are conserved, suggesting little to no net water-vapor exchange at the surface (Berkelhammer et al., 2016). Thus, for Ri measurements which fall into the super critical regime, 44% out of the 33,090 total measurements from March 2012 – June 2014, the latent heat fluxes are set to zero, providing a reminder of the significance of high stability in limiting mass transfer.

     LH is the data set most susceptible to data gaps because there must be input values of specific humidity, wind speed, and

temperature at both the 2 m and 10 m levels. Yet by using the best available meteorological data from NOAA/GMD and/or the CIBS project we estimate LH for 81% of the time period from March 2012 – June 2014. The main driver of uncertainty is the estimation of the mixing ratios with uncertainties of 53% and 30% at 2 m and 10 m, respectively, as compared to the Picarro measurements. The resultant error contribution (60%) to the LH estimate dominates the contribution from uncertainty in the wind speeds.

**2.5   Subsurface Heat Flux**

The energy flux from the overlying atmosphere to the subsurface includes direct radiative heating of the snowpack due to solar penetration (Kuipers Munneke et al., 2009), the thermal effects of wind ventilation (Albert and McGilvary, 1992), and conduction. To minimize the complications in calculating sub-surface heat flux caused by the other factors, an estimation of the conductive heat flux (C) at a depth below the solar penetration depth (at least 20 cm) combined with a heat storage (S) in

the snow above this level is used to provide an estimation of the total subsurface heat flux (G), such that

$$G = C + S. \tag{8}$$

In this study we calculate the storage heat flux across the uppermost layer and assume the heat flux to the subsurface below is equivalent to C.

     The conductive heat flux (C) represents the diffusion of heat between the subsurface and the overlying surface. The effec-

tiveness of the heat transfer is a function of the thermal conductivity of the snow (K) and the vertical temperature gradient $(\Delta T/\Delta z)$:

$$C = -K\frac{\Delta T}{\Delta z}. \tag{9}$$





The temperature gradient for the uppermost subsurface layer ($\Delta T_{01}$) is estimated as the difference between the surface temperature ($T_{surf}$, Equation 3) and the temperature measured by the shallowest, sub-surface sensor. To estimate C, at $\approx$20 cm depth, the conductive heat flux at the two levels bracketing this depth are calculated and averaged, according to the equation:

$$C = -\frac{1}{2}(K_{01}\frac{\Delta T_{01}}{\Delta z_{01}} + K_{12}\frac{\Delta T_{12}}{\Delta z_{12}}). \tag{10}$$

The thermal conductivity of the upper most layers of snow is estimated from average density profile measurements taken from five snow pits around Summit Station in July 2014. The average standard deviation of density among pits at all depths is 50 kg m$^{-3}$. There is a known annual cycle in snow density in this region based on seasonally varying thermal and snow properties (Albert and Shultz, 2002). The first two density minima with increasing depth are assumed to be different solely due to compaction of the snow over the course of a year, resulting in a linear compaction factor of -22 kg m$^{-3}$ year$^{-1}$. This factor

is used to estimate the annual evolution of near-surface snow density as a function of time from the profile measurements collected July 2014. The adjusted density profile is used to determine an average snow layer density for the representative near-surface conditions from July 2013 – June 2014. The result is a range of density values varying annually between 348 – 413 kg m$^{-3}$. Snow density is converted to thermal conductivity according to Jordan (1991) resulting in a seasonally varying thermal conductivity with an average value of 0.47 W m$^{-1}$ K$^{-1}$. The average value is higher than summer sea-ice values

(Sturm et al., 1997; Persson et al., 2002) of 0.3 W m$^{-1}$ K$^{-1}$, although the summer minimum conductivity (0.39 W m$^{-1}$ K$^{-1}$) is more similar to the sea-ice values.

The uncertainty in the conductive flux is related to the uncertainties in the calculated skin temperature, subsurface temperature, subsurface measurement height and snow conductivity estimate. The LW derived skin temperature uncertainty is approximately 0.6 K. The thermistor accuracy specifications indicate an interchangeability tolerance of 0.38 K at 0°C and

0.6 K at -40°C. We estimate the uncertainty in the measurement height of the shallowest thermistor as 2 cm. A 50 kg m$^{-3}$ uncertainty in the snow density translates to 0.1 W m$^{-1}$ K$^{-1}$ uncertainty in snow conductivity. The average temperature difference between the surface and -40 cm is about 7.2°C. The resultant uncertainty in the conductive flux, calculated by taking the quadrature sum of the fractional uncertainties is 26%.

The storage of heat in a layer is related to the time rate of temperature change averaged over that layer. The storage heat flux

(S) includes energy associated with solar heating, longwave emission, and turbulent heat flux within the snow. In the uppermost layer ($\approx$20 cm), S is calculated by the layer averaged temperature difference ($\delta T$) between chronologically adjacent time steps ($\delta t$ = 30 minutes), where $T_1$ is the temperature of the shallowest thermistor at a depth $z_1$ (simliar to Hoch (2005));

$$S = -c_{ice}\rho\left[\frac{\delta T_{surf} + \delta T_1}{2\delta t}\right](-z_1), \tag{11}$$

and $c_{ice}$ is the specific heat of ice and $\rho$ is the average density of the layer. The large uncertainty in the skin temperature

measurements (0.6°C) are close to the average temperature change from one time step to the next (0.76°C) resulting in an estimated uncertainty in S of 80%. The estimate of S is the most uncertain term in the SEB.



## 3 Results

### 3.1 Temperature (July 2013 – June 2014)

The temperature variability at and below the ice sheet surface is important for understanding the flow of heat through this interface and can influence processes like snow compaction and melt. Figure 1 depicts the variability in temperature above, below and at the surface from 1 July 2013 to 30 June 2014. The maximum surface temperature ($T_{surf}$) was -3.1°C on 10 July 2013 and the minimum was -68.8°C on 23 March 2014 (Figure 1a). A warm or cold pulse at the surface propagates to deeper portions of the GIS over time. A warm or cold event at the surface takes days to influence the temperatures at 1-2 m depth. In general, the slope of a pulse is about 10 cm of penetration per day.

In the spring, fall and winter, surface-based temperature inversions are prevalent (Miller et al., 2013) and the warmest layers of the atmosphere occur between 100-1000 m above ground level as can be seen in Figure 1a. In fact, the minimum temperature in the near-surface layer (-2 m to 20 m) occurs at the surface 46% of the year. At times the subsurface is the warmest level in the full temperature profiles (-2 m to 5 km) shown in Figure 1a. The average monthly surface temperature is colder than the average 500 m and -1 m temperatures from September to April (Figure 1b), although January 2014 had anomalously warm (compared to Januaries 2011 – 2013) surface temperatures.The maximum temperature in the near-surface layer occurs at the surface only 3.4% of the year, indicating that the default state of the system is strong surface cooling to space.

Advection of air masses over the GIS is the foundational mechanism that influences temperatures at the surface. Temperature changes at 1-5 km above ground level (AGL) are indicative of synoptic influences that transport warmer or colder air masses to Summit. During 10 January 2014 (Figure 1a) warmer air advection corresponds to relatively warm surface temperatures of -25°C. Yet there are instances, such as 15 January – 4 February 2014 with large variability in $T_{surf}$ that are not associated with large scale advection, as evidenced by fairly constant temperatures from 50 m to 5 km in altitude. The correlation between the temperatures at 5 km and the surface is 0.77 and from 1-2 km the correlation with surface temperature increases to 0.87. Seasonal synoptic variations in the free troposphere above ≈500 m influences surface temperatures, especially when the downwelling longwave emission originates from the warmest levels of the atmosphere. Synoptically driven warm air advection enhances the formation of optically thick liquid-bearing clouds, which decrease the difference in emitted longwave radiation between the air aloft and the surface. The following case studies investigate how liquid-bearing clouds effect the surface energy budget by increasing the net surface radiation.

### 3.2 Case Studies

#### 3.2.1 Liquid-bearing cloud without insolation

A case study (12 UTC 10 November to 12 UTC 11 November 2013) is used to illustrate how the different terms of the SEB interact to influence the surface temperature and surface heat exchange. Variability in this case is driven by low-level liquid-bearing clouds and the case was intentionally chosen to minimize the effects of solar influences. Cloud occurrence as measured by the MMCR up to a height of 5 km (Figure 2a) indicates a clear-sky scene at the beginning of the case study, a low-level

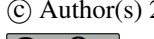



cloud from 17 UTC – 2 UTC, then a brief period of clear-sky from 2 – 3 UTC, and finally a deep cloud (> 3 km) during the end of the case study. The radar reflectivity measurements indicate the presence of ice crystals in most of these clouds. LWP values ranging from $20 – 60 \, \mathrm{g \, m^{-2}}$ from 17 - 24 UTC on 10 November (Figure 2b) show that the low-level cloud at this time is mixed phase, in contrast to the deep ice cloud at the end of the case study with little liquid present.

Coincident with the appearance of the liquid-bearing cloud, the LW↓ increased by 88 W m$^{-2}$ from 15 UTC (clear) to 23 UTC (cloudy), similar to the LW cloud radiative forcing value in Miller et al. (2015) for optically thick liquid-bearing clouds ($\approx$85 W m$^{-2}$). This cloud radiative effect resulted in an increase in $T_{surf}$ and thus LW↑ of 43 W m$^{-2}$ (Figure 2c). During the clear-sky period the boundary-layer was weakly stable (Ri = 0.15), but the occurrence of the liquid-bearing cloud and its warming effect on the surface changed the stability to neutral (Ri $\approx$ 0), (Figure 2d). During the transition back to clear-sky

(2 UTC), LW↓ decreased by about 70 W m$^{-2}$ and the Richardson number became critically stable. LW↓ was smaller in the presence of the deep ice cloud, compared to the liquid-bearing cloud, resulting in a much smaller LW↑ at the time. In the presence of the deep ice cloud the boundary-layer became weakly stable again (Ri = 0.2).

Changes to the net radiative flux caused by the cloud (Figure 2e) elicited a response in the other SEB terms. On 10 November from 15 UTC to 23 UTC the sensible heat flux decreased by a factor of 2, from 36 to 18 W m$^{-2}$. The conductive heat flux

changed from having a warming effect on the surface by +8.1 W m$^{-2}$ to having a -0.3 W m$^{-2}$ cooling effect by 23 UTC. The average latent heat flux increased from 0.8 W m$^{-2}$ during the clear-sky period (12 – 18 UTC) to an average value of 2.4 W m$^{-2}$ during the cloudy period (18 – 24). The net result is that the liquid-bearing cloud increased the surface temperature from -47.8°C (15 UTC) to -33.0 °C (23 UTC). This is half the temperature increase that would have occurred ($\approx$28.4°C) if the entire LW↓ increase (88 W m$^{-2}$) had gone toward heating the surface. This example demonstrates how changes to the

turbulent and conductive heat fluxes are an important compensation mechanism that modulates surface warming due to cloud radiative forcing. This damping effect on the radiative forcing by the response terms was noted by previous Arctic researchers (e.g., Persson, 2012; Sterk et al., 2013; Solomon et al., 2016).

The subsurface cooled in response to the surface cooling during the clear-sky period on the 10th (Figure 2f), yet the minimum measured temperature at -0.2 m (-41.8 °C) was not realized until 18 UTC. This shallowest subsurface temperature sensor (-0.2

m) cooled by 0.8°C from 12 UTC to 18 UTC on 10 November. The cooling from above at -0.2 m on 10 November was damped by the relatively warm snowpack below. During the liquid-bearing cloud period the subsurface layer at -0.2 m was warmed from above and below allowing for a 1.8°C temperature increase from 18 UTC 10 November to 2.5 UTC on 11 November. This suggests that a time lag of the effect of the surface temperature on the subsurface temperatures is important in determining the ground heat flux. The heat storage in the upper layer of snow had an average value of -12.9 W m$^{-2}$ for the 24 period shown

in Figure 2e, indicating that a portion of the increase in LW↓ went toward increasing the internal energy of the top layer of snow. Large negative values of S occur during the transition from clear to the onset of the liquid-bearing cloud presence (17 - 20 UTC), as this layer warms rapidly, and visa-versa during the transition back to a clear-sky scene (0 - 2 UTC).



### 3.2.2 Liquid-bearing cloud with insolation

A case study on 6 August 2013 also illustrates the longwave warming effect of liquid-bearing clouds and investigates the additional influence of shortwave radiation. Similar to the first case study, surface temperature variability is driven by the downwelling radiative flux, which in this case is a combination of longwave and shortwave influences.

MMCR measurements (Figure 3a) indicate a clear-sky scene from 2 to 6 UTC, a low-level cloud from 6 - 13.5 UTC, clear-sky from 13.5 to 16 UTC, a deep cloud from 18 to 22 UTC, and finally a low-level cloud during the last hour of the case study period. The low-level cloud is mixed phase from 6 to 13.5 UTC and LWP values ranging from $0 - 15 \mathrm{\ g\ m^{-2}}$ (Figure 3b) indicate that it is optically thin. LWP values ranging from $0 - 20 \mathrm{\ g\ m^{-2}}$ also indicate that the deep cloud later in the day is mixed phase from 18 to 21 UTC although after $\approx$19 UTC LWP values are low due to competition from falling ice into the

mixed phase layer from above. The low-level cloud from 23 to 24 UTC is optically thicker then the previous low-level cloud with LWP ranging from $5 - 30 \mathrm{\ g\ m^{-2}}$.

    The presence of the optically thin liquid-bearing cloud (6 – 13.5 UTC) produces an approximate increase of $70 \mathrm{\ W\ m^{-2}}$ of LW↓ compared to the preceding clear-sky scene. Over this period shortwave radiation increases the net radiation at the surface by an additional $5 - 75 \mathrm{\ W\ m^{-2}}$. In response, LW↑ radiation increases by $50 \mathrm{\ W\ m^{-2}}$. The combination of thin liquid-bearing

clouds and insolation produces positive net radiation at the surface from 9.5 to 13 UTC (Figure 3c). During the daytime clear-sky period the net radiation is near zero, indicating that shortwave warming is offset by the longwave cooling at the surface. Net radiation again goes positive in the presence of liquid-bearing clouds that occur after 18 UTC. After 18 UTC the net radiation declines as the solar radiation diminishes.

    The compensating response of the non-radiative terms to changes in the downwelling radiation, shortwave and/or longwave,

are similar to the November case study. The sensible heat flux decreases from $29 \mathrm{\ W\ m^{-2}}$ at 5 UTC to -9 at 12.5 UTC (Figure 3d). The fact that the SH is negative during the presence of the liquid-bearing cloud indicates that the surface temperature is warmer than the 2 m temperature, thus the near-surface atmospheric layer is unstable. The conductive heat flux decreases from $9 \mathrm{\ W\ m^{-2}}$ at 5 UTC to $0 \mathrm{\ W\ m^{-2}}$ at 12.5 UTC, indicating the subsurface temperature gradient as been reduced (Figure 3e). The 10 m temperatures from 9 to 16 UTC are questionable thus LH is not shown during this period. During the daytime clear-sky

period (13.5 to 16 UTC) the net radiation is near zero as is the ground heat flux and sensible heat flux. In the presence of the deep mixed-phase cloud after solar noon the net radiation again is positive, the sensible and ground heat flux are near zero, and the latent heat flux is approximately -10 W m$^{-2}$.

### 3.3 Surface Energy Budget

### 3.3.1 Annual Cycle

Scaling up the analysis to the full annual cycle (July 2013 – June 2014), monthly averages of the four SEB terms from Equation 1 illustrate the seasonal balance of energy fluxes at the surface (Figure 4). The bottom numbers in Figure 4 indicate the percentage of the month for which all 4 SEB terms are available. In addition, Figure 4 includes additional data for Q, SH, and LH indicating that July 2013 – June 2014 is, in general, consistent with previous years and indicates that January 2014 was





somewhat anomalous. The extended data periods for Q, SH and LH all end June 2014 and include start dates of January 2011, June 2011, and March 2012, respectively.

The sensible and radiative heat fluxes have nearly compensating influences on the SEB during the non-summer months when temperature inversions are prevalent. During the summer, on average, all SEB terms are relatively small in magnitude.

The monthly mean total radiative flux (Q) is positive in June and July (Figure 4). Only these two months correspond to periods when the amount of absorbed SW exceeds the net LW radiational cooling. June and July are also when the sensible and latent heat fluxes are at their seasonal minima. The subsurface heat flux monthly minimum values occur a month earlier in the year, due to the cooler subsurface temperatures in the spring compared to the fall (Figure 1). Colder subsurface temperatures enhance the ability of the GIS to remove heat from the surface via conduction, resulting in a mean cooling effect in the spring

and warming effect in the fall.

Over the entire year the SEB residual, or the sum of all the SEB terms, when available (75.3% of the time), is 0.9 W m$^{-2}$. The monthly residuals (top numbers in Figure 4) indicate that there are times of the year when the residuals are larger but there is no apparent seasonality in the combined SEB terms. Generally, the monthly mean residuals could be due to energy imbalances, under sampling, measurement biases, and/or measurement uncertainties. Each monthly residual is below the total

SEB uncertainty (excluding the S term) of 12.4 W m$^{-2}$.

### 3.3.2 Diurnal Cycles

The magnitudes of the monthly mean SEB terms are small from May – August (< 10 W m$^{-2}$), yet the diurnal variability peaks during this period, driven largely by the solar cycle. The net radiative flux increases during times of peak insolation (Figure 5a), although the high surface albedo limits the maximum Q to 40 W m$^{-2}$. The maximum values of the net radiative flux occur in

July, when the sun still rises more than 30° above the horizon and liquid-bearing clouds are frequent (Figure 6a, b), which act to radiatively warm the surface at Summit Station year round (Miller et al., 2015).

Counteracting the net radiative flux, the sensible heat flux is negative for large sun angles and warms the surface by approximately 20 W m$^{-2}$ when the sun is below the horizon (Figure 5b). The diurnal variability for this term is largest in summer due to an enhanced diurnal cycle of the near-surface temperature gradient (Miller et al., 2013). The cooling effect of the conductive

heat flux (Figure 5c) is most prominent when the sun is above the horizon and is maximized at solar noon. In agreement with the results in Figure 4, more conductive surface cooling occurs in the spring compared to the fall due to the lag in subsurface response, which results in relatively colder subsurface temperatures in the spring. The diurnal variability of the latent heat flux is largest in June ranging from hourly-average values of -33 to 12 W m$^{-2}$ (Figure 5d) due to an increase in available moisture (Figure 6c).

Sun angle, and the associated change to the net radiative flux, is a main driver of energy fluxes at the surface (Figure 5). The monthly-hourly energy fluxes in panels b-d are generally anti-correlated with the net radiative flux in panel a (correlation coefficients are: b = -0.81, c = -0.65, d = -0.69). The responses of the latent, sensible and conductive heat flux terms to changes in $LW \downarrow + SW_{net}$ are investigated in Section 3.4.



### 3.4 Responses to Surface Radiative Forcing

The surface energy budget at Summit Station is largely driven by changes in the downwelling radiation. In general, the LW↑, turbulent, latent, and subsurface heat fluxes (response terms) respond to changes in the LW↓ and net SW flux (forcing terms). The response terms are not always governed by the forcing terms, as, for instance, under high wind conditions the turbulent

heat fluxes can operate independently as the Ri in these cases is dominated by the wind shear (see Equation 7). Cloud presence influences the radiational balance at the surface by modulating the downwelling radiation; increasing LW↓ and decreasing SW↓. Miller et al. (2015) show that clouds increase the net surface radiation compared to an equivalent clear-sky scene, because the high year-round surface albedo limits the magnitude of the cloud SW cooling effect to less than that of the LW warming effect. Statistical relationships for the current study reinforce the fact that liquid-bearing clouds increase the forcing

terms during two distinct periods; with and without solar insolation (Figure 7a). Hence, the occurrence of liquid-bearing clouds correspond to warmer surface temperatures in both circumstances (Figure 7b) and consequently greater LW↑ (Figure 7c), which is proportional to the surface temperature to the fourth power.

LW↑ has less variability (all cases in Figure 7c) than the variability of the forcing terms (all cases in Figure 7a). In addition, the differences between the cloudy and non-cloudy states are more pronounced in Figure 7a, compared to Figure 7c. Thus,

compensation by the non-radiative SEB terms must account for imbalances to the radiative flux at the surface, as illustrated in the case studies presented in Section 3.2 and in Figures 4 and 5. The annual cycle of the responses of LH, SH, G and LW↑ are explored in Section 3.4.2 after investigating the effect of liquid-bearing clouds and/or sun angle on boundary-layer stability (Section 3.4.1).

### 3.4.1 Boundary-Layer Stability Response

The degree to which the overlying atmosphere can dynamically interact with the surface is important for determining the turbulent heat exchange. Atmosphere/ice sheet interaction is modulated by low-level stability, which can be influenced by both thermodynamic and dynamic processes. Mechanical mixing, via high wind speeds, is one way to decrease near-surface stability and increase turbulence near the surface. The 10 m wind speed is greater than $8 \text{ ms}^{-1}$ for 16% of 32130 stability estimates. The median Richardson number decreases from 0.19 for all cases to 0.06 for the cases that report higher wind speeds (>8

$\text{ms}^{-1}$), showing the expected decrease of stability. In addition, cloud-driven atmospheric mixing can also affect the low-level atmospheric structure (Shupe et al., 2013a) and liquid-bearing cloud presence, especially in combination with enhanced solar radiation, decrease the near-surface temperature gradient (Hudson and Brandt, 2005; Miller et al., 2013).

This study explicitly shows that the radiative influences of liquid-bearing clouds and/or insolation create neutral or unstable boundary-layer conditions. When the sun is below the horizon, as for the first case study (Section 3.2.1), the presence of liquid-

bearing clouds decreases the stability such that a majority of the cases are weakly stable (0 < Ri < 0.25) (Figure 7d). In the absence of liquid-bearing clouds (LWP < 5 $\text{gm}^{-2}$) the surface radiatively cools, the stability increases, and consequently a majority of the cases are strongly stable (Ri > 0.25). Solar radiation (SZA < 70°) warms the surface sufficiently to decrease the near-surface stability (Figure 7d). When the sun is present yet there are no liquid-bearing clouds the median Ri is weakly





stable. However, when optically thick liquid-bearing clouds (LWP > 30 $\mathrm{gm^{-2}}$) are present the boundary-layer is near-neutral on average. Interestingly, optically thin liquid-bearing clouds (5 $\mathrm{gm^{-2}}$ > LWP > 30 $\mathrm{gm^{-2}}$) lead to more frequent occurrence of more unstable conditions in the presence of insolation, because these clouds emit significant longwave radiation while also allowing significant penetration of solar radiation, thus producing the maximum surface heating. Our results that liquid-bearing

clouds of intermediate thickness lead to higher instability agree with studies that show these clouds produce the maximum cloud radiative forcing for elevated sun angles (Bennartz et al., 2013; Miller et al., 2015). Hence, liquid-bearing clouds and/or solar insolation enhance turbulent mixing, facilitating sensible and latent heat exchange, although instability (negative Ri) requires SW↓.

### 3.4.2 SEB Responses

Process-based relationships distill our understanding of the underlying physical processes into a succinct form that is informative, yet practical. While clouds, the solar cycle, and other processes can influence the downwelling radiation, process relationships between response terms and forcing terms reveal how variability in downwelling radiation affects the other SEB terms. A linear regression of the relationship between the forcing and response terms yields a slope of -1.01 (Figure 8a), indicating that the SEB is largely radiatively driven, the response terms account for all of the forcing energy flux, and there is

approximate closure for the SEB terms calculated here. The scatter in this relationship is due to measurement uncertainties, mismatches of response times in different terms, and the spatial distribution of the instrumentation. The annual evolution of this slope (Figure 9) shows that the SEB response terms balance the forcing terms to within ≈10% in all months of the year. Thus, any change in forcing terms elicits an approximately equal change in flux through the combination of response terms.

The response to the radiative forcing can be evaluated for each term independently (Figure 8b-e), and as a function of

month, showing that each term responds differently throughout the annual cycle (Figure 9). The slope of the linear fit provides an estimate of the relative magnitude (percentage) of the response of each term. The RMS error of the monthly response estimates in Figure 9 are calculated by comparing the estimated values, using the linear fit, to the measured values (Figure 10). The RMS error includes the uncertainty of the measurements involved, any delay in response time greater than 30 minutes, and variability in the physical response not represented by the linear fit. Generally, the RMS error of the linear fits of all response

terms to the driving terms are on the same order of magnitude as the combined uncertainty of the SEB components.

The annual response in the LW↑ term (Figure 8b) is the largest (0.70) out of all the response terms, as its magnitude is directly proportional to the surface temperature to the fourth power. The annual cycle of this response shows a weaker response in summer (40-50%) and a stronger response in winter (55-75%). The lower response of the LW↑ term in June 2014, compared to winter months during December 2013 –February 2014 (or compared to values from June 2011 – 2014) is partially due

to the increased response of the latent heat flux for this specific month (Figure 9). Any increase (decrease) of response of an individual term will effectively decrease (increase) the change in surface temperature, and hence the response of LW↑, to radiative forcing.

The response of the sensible heat flux is fairly constant at ≈ -0.11 throughout the annual cycle (Figure 9) due to its dependence on both the near-surface temperature gradient and stability (heat transfer coefficient – see Equation 5). For weakly





stable conditions, the former term dominates decreasing (increasing) the heat flux for surface warming (cooling), while for very stable conditions the latter term dominates limiting turbulent exchange and increasing (decreasing) the sensible heat flux for surface warming (cooling) (e.g., Grachev et al., 2005). Since this Summit data generally shows a decrease in sensible heat flux for an increase in the forcing terms (surface warming), this is consistent with weakly stable conditions on the unstable side

of the stability transition shown by Grachev et al. (2005). Therefore, the response of the sensible heat flux to changes in the surface temperature is similar throughout the year and does not show an annual cycle. However, the RMS error of the linear fit (Figure 10) during winter (9.7 $\mathrm{Wm^{-2}}$) is greater than during summer (6.0 $\mathrm{Wm^{-2}}$) (i.e., there is more scatter in the sensible heat response in winter), suggesting that conditions in winter are at times very stable and that the sensible heat flux response to radiative forcing is then different. In summer, conditions are rarely very stable so the response in sensible heat flux is more

strongly correlated with the change in the forcing terms.

The response of the latent heat flux increases in summer compared to other months of the year (Figure 9). The amount of available moisture (Figure 6c) peaks in summer and average PWV values for non-summer (winter) months are below 2 mm (1 mm). Thus, changes to near-surface stability due to changes in the forcing terms produce a smaller response when moisture gradients are small in magnitude.

The response of the conductive heat flux to radiative forcing is greatest in winter (Dec – Feb) at 22% compared to ≈10% in summer (June- August). Seasonal changes in the conductive heat response are due to changes in snow density, thermal conductivity, and subsurface temperatures. Warmer subsurface temperatures resulting from prior warm surface temperatures precondition the snowpack, reducing its ability to remove heat from the surface. Decreased density in the summer decreases the thermal conductivity of the near-surface snow pack, also limiting the ability of the subsurface to remove energy from the

surface. The RMS error of the linear fit of the conductive heat flux to the forcing terms is relatively low with an annual mean of 3.2 $\mathrm{Wm^{-2}}$.

The response of the heat storage in the upper subsurface layer is important to consider when accounting for all the energy responses at the surface. Even though the annual mean of S is less than 1 $\mathrm{Wm^{-2}}$ (i.e., there is effectively no annual net change in temperature in the near-surface snow), it is highly variable (annual standard deviation = 62.5 $\mathrm{Wm^{-2}}$) as this layer can

warm or cool rapidly from one half hour period to the next. The heat storage response to the forcing terms also accounts for subsurface heating due to solar penetration. Over the annual cycle the response of S is 7%. The response of S is at a maximum in April – May at 23-25% (Figure 9), indicating relatively cold near-surface snow is able to store larger amounts of energy originating from radiative sources.

Since scatter in S in response to forcing is so large, we first examine the scatter of all the other terms jointly. The RMS error

of the linear fit of (LH + SH + C - LW↑) vs. (LW↓ + SWnet) is maximum in July (19.6 $\mathrm{Wm^{-2}}$) and has an annual mean value of 14.9 $\mathrm{Wm^{-2}}$ (Figure 10). The maximum RMS error occurs in summer due to an increase in the latent heat RMS error of the linear fit from an annual average value of 8.9 $\mathrm{Wm^{-2}}$ to 15.3 $\mathrm{Wm^{-2}}$ in summer. The RMS error of the linear fit of S is lowest in January (36 $\mathrm{Wm^{-2}}$) and highest in August (89 $\mathrm{Wm^{-2}}$) and has monthly mean RMS error of 59 $\mathrm{Wm^{-2}}$. The high variability, uncertainty, and generally weaker relationship of S with the forcing terms indicate that the estimation of S is the

largest unknown in closing the energy budget on short time scales. The 1-sigma uncertainty of the response of LH + SH + C +




S - LW↑ to the forcing terms, shown by the error bars in Figure 9, is primarily due to the variability and associated uncertainty in S. However, correctly accounting for the ground heat flux in the upper most layer provides near closure of the surface energy balance, a critical accomplishment of the synthesis of comprehensive datasets given here.

At the ice sheet/atmosphere interface surface temperature is the linchpin that connects the subsurface to the atmospheric boundary layer, responding to changes in the net flux at the surface. The variability in the surface temperature is controlled by changes in the forcing terms and modulated by the response terms. An increase in radiative forcing warms the snowpack; increasing the surface temperature and decreasing the near-surface atmospheric stability. Not surprisingly, the response terms are all associated with surface temperature; either directly proportional, or a function of the near-surface temperature gradient. Latent heat flux is also dependent on the near-surface moisture gradient and the ground heat flux is dependent on the thermal conductivity of the snow pack leading to seasonal differences in their responses. This study highlights the importance of the seasonal changes in the non-radiative responses, which determine the annual cycle of the LW↑ response.

The seasonal response of the SEB to cloud presence is estimated by combining the radiative effects of clouds with the observationally based and statistically derived relationships between the forcing and response terms. Cloud radiative forcing (CRF) at the surface, as detailed in Miller et al. (2015), is the instantaneous net radiative effect of clouds. Furthermore, changes in the forcing terms elicit a response of the surface temperature and the non-radiative SEB terms. Thus, we combine the monthly CRF values reported in Miller et al. (2015) and monthly responses, calculated from the maximum available data (Figure 9), to estimate the corresponding increase in LW↑ and decreases in SH, LH and G attributed to cloud presence. Figure 11a shows LW↑ has the smallest increase due to CRF in May (9.8 $\mathrm{Wm^{-2}}$), the largest increase in October (30.5 $\mathrm{Wm^{-2}}$), and an annual mean response of 20.7 $\mathrm{Wm^{-2}}$. The non-radiative responses to the annual CRF value of 32.9 $\mathrm{Wm^{-2}}$ are -2.9 (SH), -0.7 (LH), and -9.6 (G) $\mathrm{Wm^{-2}}$. Subtracting the monthly LW↑ response from the monthly mean LW↑ yields an estimate for the amount of LW radiation that would be emitted by the GIS surface in the absence of clouds. Comparing the monthly mean surface temperatures, derived from the measured LW↑ and the estimated clear-sky LW↑, produces the approximate monthly differences shown in Figure 11b, suggesting that clouds increase the surface temperature by 6.9°C annually during the time period January 2011- October 2013.

## 4  Summary

Characterization of surface energy fluxes and their variability illuminates the important processes that control surface temperatures in central Greenland. Here observations from Summit Station are used to derive all terms of the surface energy budget and to examine key relationships among these terms and with other key atmospheric drivers. Despite the harsh Arctic environment SEB estimates could be made for all the terms for 75% of the year spanning July 2013 – June 2014.

Over the annual cycle atmospheric temperatures in the free troposphere (> 1 km) are well correlated with surface temperatures, although energy exchange processes at the surface enhance surface temperature variability. In general, time-series data, monthly mean values, and monthly diurnal cycles all show that the non-radiative SEB terms oppose the increase or decrease of the net radiation. Liquid-bearing clouds and solar insolation strongly modulate the radiative flux that reaches the surface,





which affects subsurface temperatures, the stability of the boundary-layer, and the near-surface temperature gradients. A pair of case studies illustrate how all the pieces fit together to depict how an increase in surface radiation elicits a response in the surface temperature, while also indicating that the increase in temperature is lessened by a decrease in sensible and conductive heat fluxes. The resultant compensation of the non-radiative SEB terms thereafter affects the net amount of surface warming that occurs due to cloud radiative forcing and/or insolation. Similar compensation is apparent when looking at longer-term averages.

To examine these relationships in more detail, radiative forcing terms (LW↓ + net SW) were related to the response terms (SH, LH, C, S and -LW↑) throughout the annual cycle. Linear regression analysis, for the year-round dataset relating the response terms as a function of the forcing terms, resulted in a -1.01 slope, indicating general closure in the calculated SEB terms. On average LW↑, which is directly linked to surface temperature, responds by about 70% of a perturbation in incident radiation, with a somewhat diminished response in summer. Quantifying how each non-radiative response changes throughout the year provides insight into how much SH, LH and/or G limit the surface temperature increase due to the occurrence of liquid-bearing clouds and/or insolation:

– Latent heat flux response is near-zero for much of the year, with an increased response in summer.

– Sensible heat flux response is fairly constant throughout the annual cycle (11%).

– Ground heat flux, consisting of both heat storage in the upper most ≈ 20 cm of snow and the conductive flux below this layer, is the largest non-radiative response for most of the year, with a decreased response in summer.

The enhanced summer latent heat flux response is due to an increase in available moisture and an increase in turbulence during relatively frequent periods of neutral/unstable near-surface conditions. In winter the effect of the stable boundary-layer is to dampen the response of the turbulent sensible heat flux, yet this dampening effect is offset by the enhanced near-surface temperature gradient. The consequence of a limited sensible heat exchange during periods of strong radiational cooling is that the sensible heat flux response is relatively constant throughout the annual cycle. Finally, the ground heat flux response decreases in the summer due to decreases in near-surface snow density and warmer subsurface temperatures.

Previous studies by Cullen (2003) (July 2000 – June 2001) and Hoch (2005) (June 2001 – July 2002) also report the annual cycle of the surface energy budget components at Summit Station. Other than the fact that January 2014 did not follow the annual cycle of the SEB variations, due to the atmosphere being relatively warm and cloudy, the monthly mean values of this study in the winter are similar to the earlier studies. However, the summer monthly total radiation is lower in our annual cycle (July 2013 - June 2014) compared to previous studies, thus the recent monthly averages for non-radiative terms are greater than the 2000 values. The differences in the annual cycles could be due to possible decreases in cloud cover (Comiso and Hall, 2014).

In central Greenland, cloud presence in winter (longwave forcing) is unable to produce a neutral stratification. It is only with insolation that neutral and unstable conditions exist. In contrast, over Arctic sea ice, wintertime conditions are near-neutral or even slightly unstable nearly 25% of the time (Persson et al., 2002). More instability over sea ice compared to the Greenland





ice sheet may be due to warming of the surface from below due to oceanic influences. Springtime/summertime near-neutral and slightly unstable conditions with shortwave forcing observed here is similar to that observed over sea ice (e.g., Ruffieux et al., 1995; Persson et al., 1997, 2002). Also in agreement with our findings are process diagrams obtained via a modeling study over sea ice (Sterk et al., 2013) that found the non-radiative SEB terms lessen the change in surface temperature due to changes

in downwelling radiation. Moreover, observational studies over sea-ice (Persson et al., 2002) and in the Greenland ablation zone (van den Broeke et al., 2011) suggest that if/when Summit Station more frequently experiences melt the non-radiative compensation, detailed in this study, may be significantly diminished as energy goes towards surface melting.

These central Greenland results can be used to evaluate how well the annual and diurnal cycles of the SEB terms are represented in climate models and reanalyses, and specifically the relationship among key terms. It is known that global

climate models underestimate the occurrence of liquid-bearing clouds above Greenland (Kay et al., 2016). We estimate that the underrepresentation of clouds, especially liquid-bearing clouds, should produce annual surface temperature biases ranging from 0 to -6.9°C. If the representation of liquid-bearing clouds were to improve then the modeled downwelling radiation would likely also improve, but it is unclear if the other SEB terms would realistically adjust. A regional or global climate model's modus operandi is to achieve absolute closure of the SEB, hence this study will be useful in future studies as a valuable tool

for pinpointing the processes responsible for possible model surface temperature bias over Greenland and for evaluating model representation of physical processes at the ice sheet/atmosphere interface.

*Acknowledgements.* This research is supported by the National Science Foundation under grants PLR1303879, 1314156, and 1023574. Thanks to the Swiss Federal Institute for providing the ETH broadband radiometer measurements. Near-surface meteorological tower data are provided by the National Oceanic and Atmospheric Administration's Global Monitoring Division. Sensible heat flux calculations via the

eddy correlation method are made using the code written by Andrey Grachev for Tiksi, Russia. Thanks to the various science technicians and Polar Field Services for their excellent support of the field experiments at Summit Station.



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

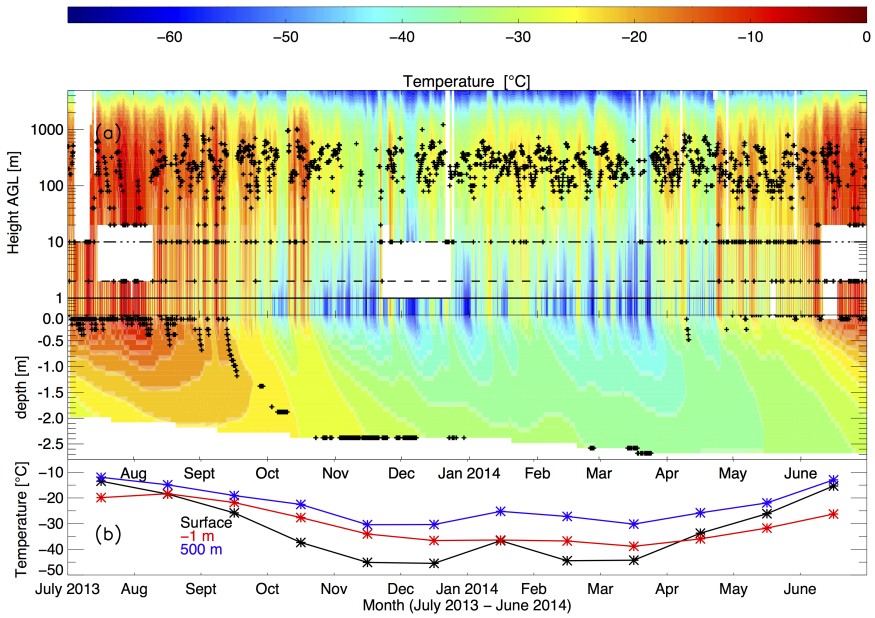

**Figure 1.** Temperature evolution from 1 July 2013 – 30 June 2014. (a) Values between the solid horizontal lines indicate surface temperatures ($T_{surf}$). The dashed (dashed-dotted) line at 2 m (10 m) level is NOAA/GMD measurements, and from 20 m to 5km above ground level (AGL) is derived from twice-daily radiosoundings. The height scale AGL is logarithmic to emphasize the near-surface values where the atmospheric and GIS are physically coupled. Subsurface temperatures are on a linear scale. White areas indicate periods of data gaps and black symbols indicate the height of the maximum temperature in each profile. (b) Monthly mean temperatures at 500 m, $T_{surf}$ and -1 m.





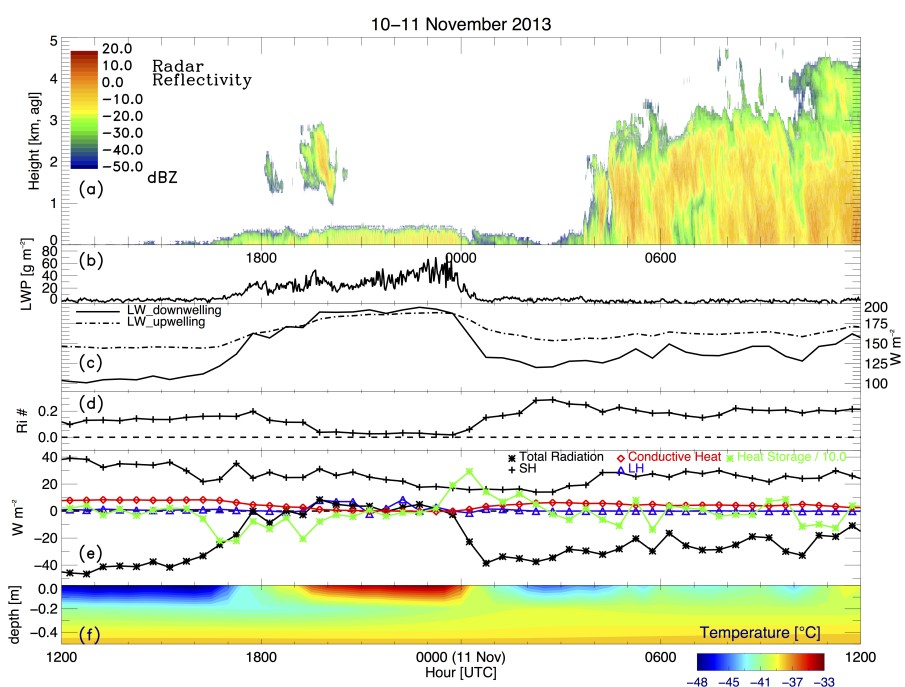

**Figure 2.** A case study from 12 UTC on 10 November 2013 to 12 UTC on 11 November 2013. (a) Cloud occurrence as seen by the MMCR, (b) liquid water path, (c) longwave upwelling and downwelling radiation, (d) Richardson number, (e) surface energy fluxes: total radiation, sensible heat, latent heat, conductive heat and heat storage/10.0 and (f) subsurface temperatures.





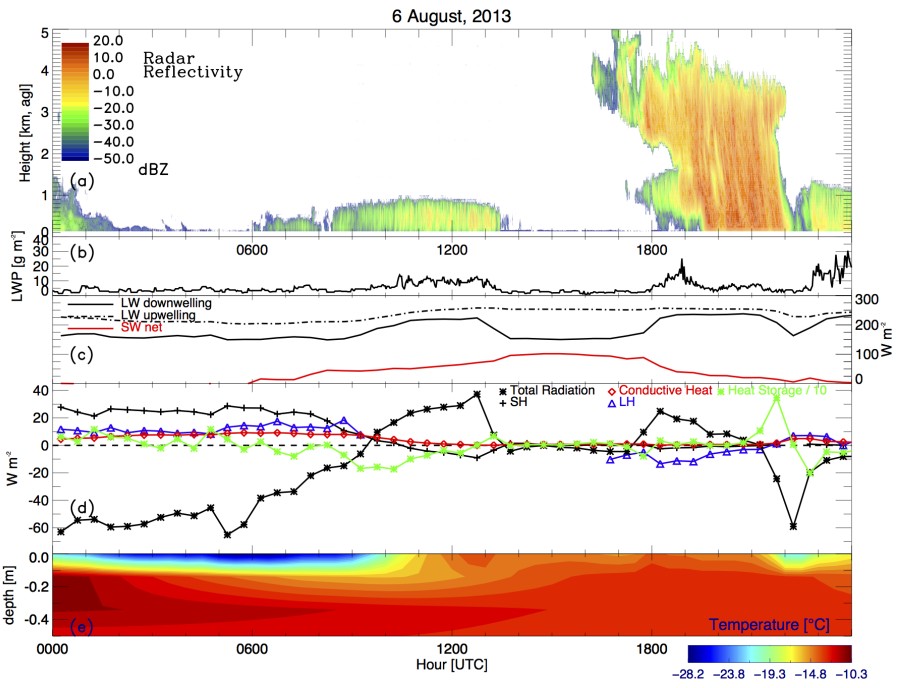

**Figure 3.** A case study on 6 August. (a) Cloud occurrence as seen by the MMCR, (b) liquid water path, (c) longwave upwelling, longwave downwelling, and net shortwave radiation, (d) surface energy fluxes: total radiation, sensible heat, conductive heat and heat storage/10.0 and (e) subsurface temperatures.





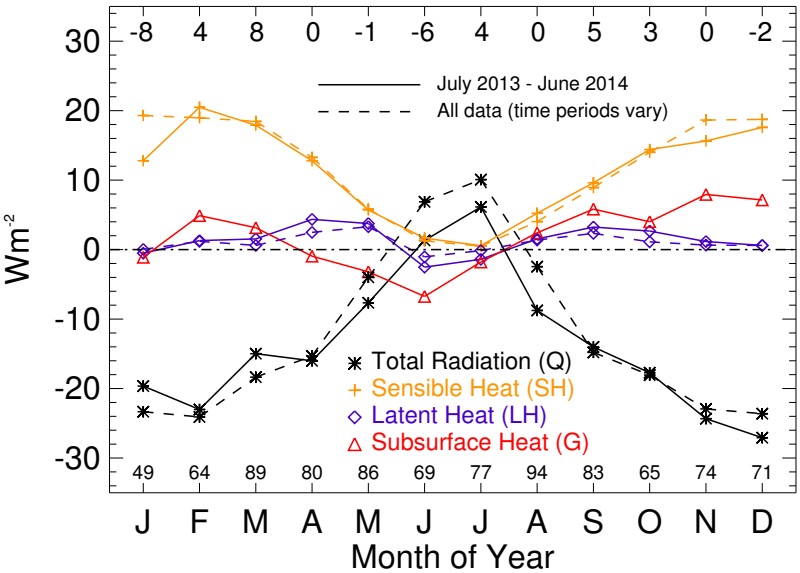

**Figure 4.** Monthly mean values of the 4 SEB terms for the period July 2013 – June 2014. The values at the top of the figure are the monthly residual of the SEB (W m$^{-2}$). The values at the bottom of the figure are the percentage of the month for which all 4 SEB terms are available.

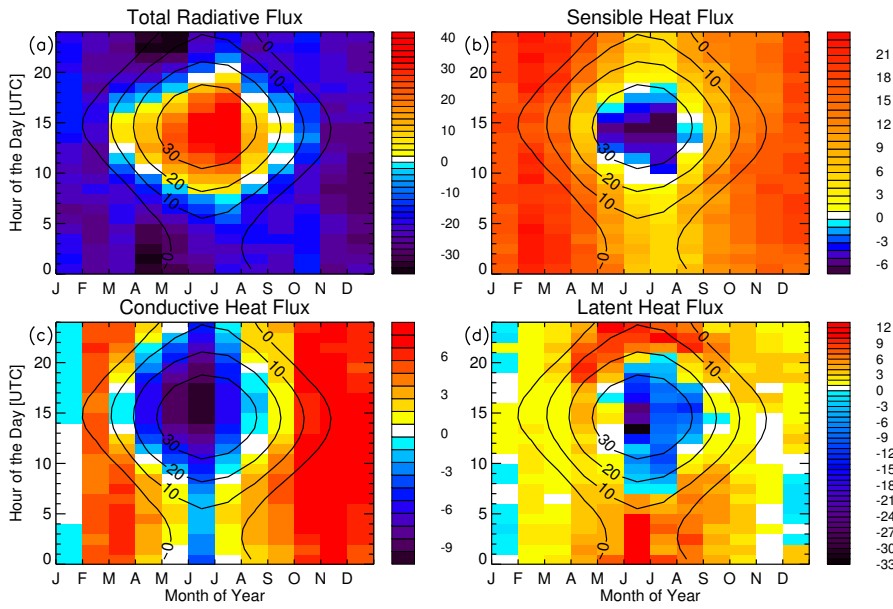

**Figure 5.** Monthly-hourly mean values from July 2013 – June 2014 of (a) total radiative flux, (b) sensible heat flux, (c) conductive heat flux and (d) latent heat flux. Black contour lines indicate the solar elevation angle. Units on the color bars are all in W m$^{-2}$.





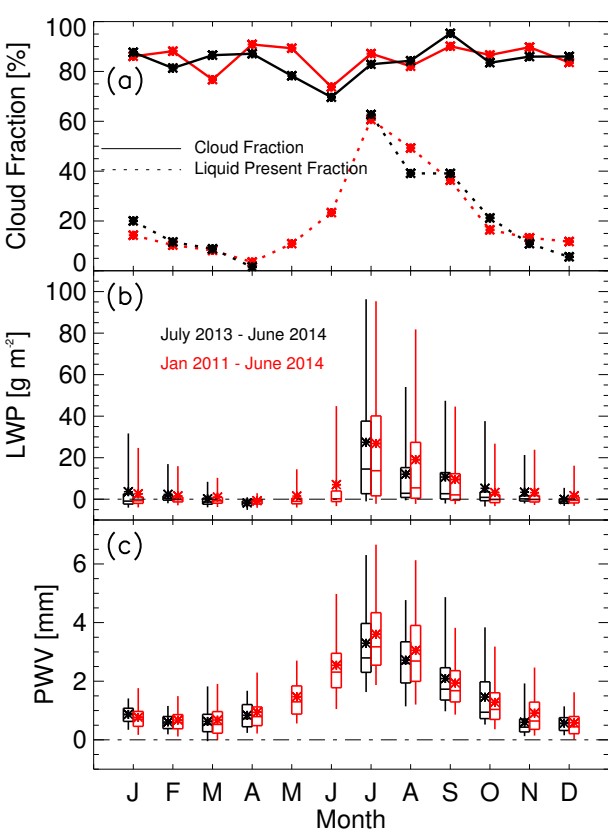

**Figure 6.** a) Cloud fraction (solid) and liquid present fraction (dotted, LWP > 5 g m$^{-2}$), b) liquid water path and c) precipitable water vapor. Statistics shown in black (red) are for available data spanning July 2013 – June 2014 (January 2011 – June 2014). Distributions are represented by box-and-whisker plots (the box indicates the 25th and 75th percentiles, the whiskers indicate 5th and 95th percentiles, the middle line is the median, and the * is the mean).





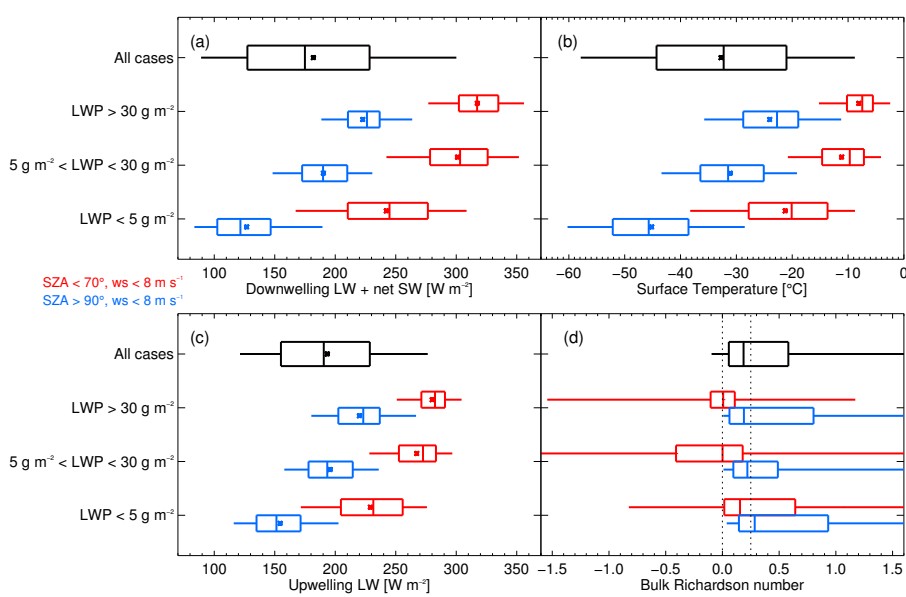

**Figure 7.** Statistics of (a) LW↓ + net SW, (b) surface temperature, and (c) LW↑ for the period spanning January 2011 - June 2014. (d) Statistics of the bulk Richardson number for the period spanning March 2012 – June 2014. The black distribution represents all quality controlled cases. The red (blue) distributions represent periods when the windspeed < 8 m s$^{-1}$ and the solar zenith angle is < 70° (> 90°). Distributions are represented by box-and-whisker plots (the box indicates the 25th and 75th percentiles, the whiskers indicate 5th and 95th percentiles, the middle line is the median, and the * is the mean).





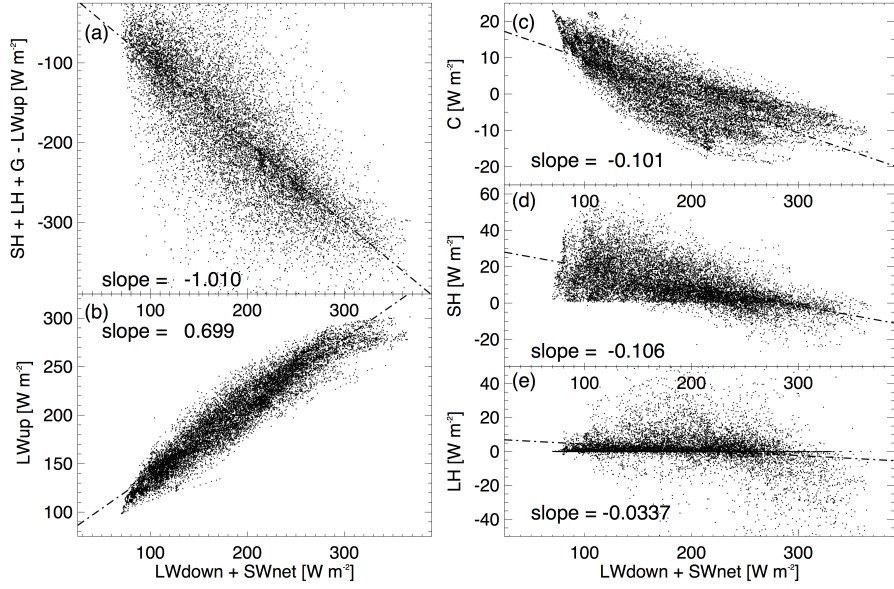

**Figure 8.** Linear regression of data from July 2013 – June 2014. (a) Total response (SH, LH, -LW↑, and G) as a function of the forcing terms (LW↓ + net SW). (b) LW↑, (c) conductive heat, (d) sensible heat, and (e) latent heat flux as a function of the forcing terms. The slope of the best fit linear regression is included in each panel.

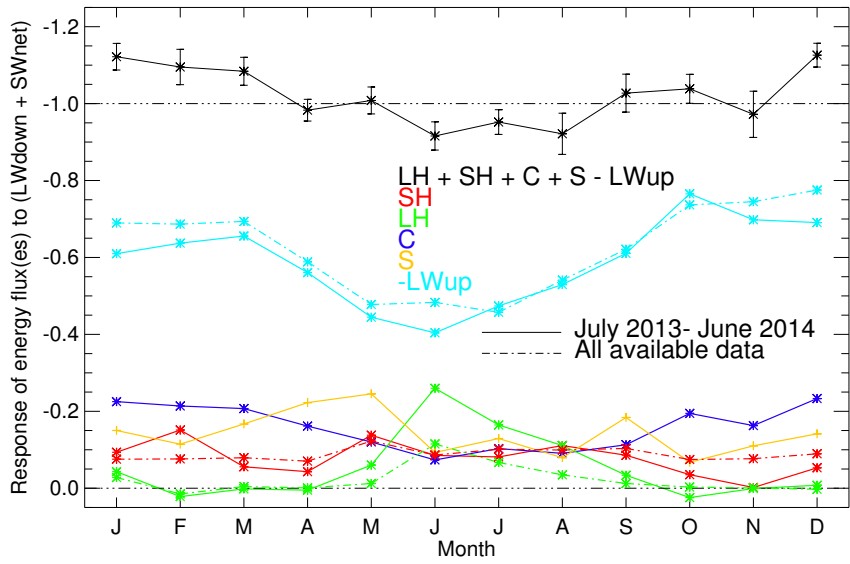

**Figure 9.** Annual cycle of monthly linear regression of responses to the forcing terms. The solid lines are for data spanning July 2013 – June 2014 during which all SEB estimates are available. The dashed lines are representative of all available data for the given subset. Note that the y-axis decreases upwards.





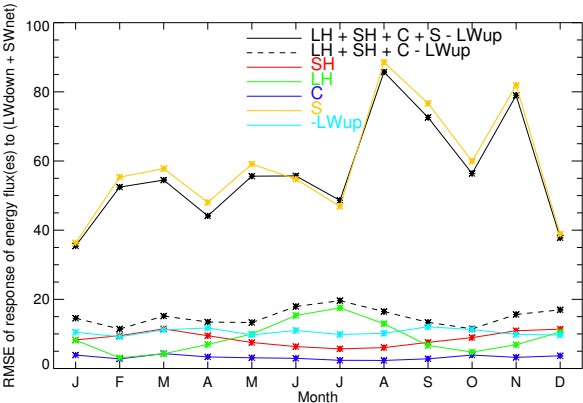

**Figure 10.** Root Mean Square Error ($\mathrm{W\ m^{-2}}$) computed from the differences between the measured response of a given term (or combination of terms) and the estimated monthly responses in Figure 9.

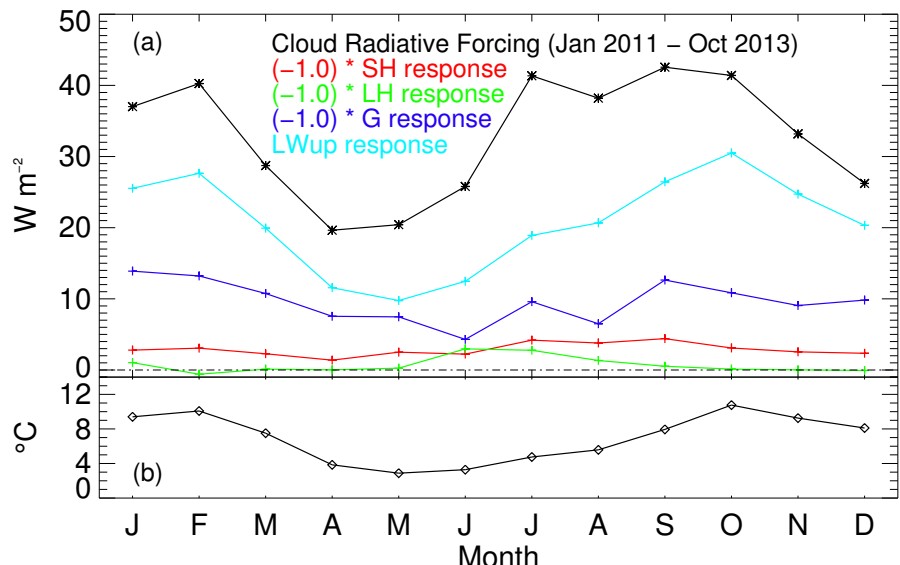

**Figure 11.** (a) The annual cycle of cloud radiative forcing (black) from January 2011 - October 2013 (Miller et al., 2015) and estimated annual cycle of responses, calculated from the values in Figure 9, of sensible heat flux, latent heat flux, ground heat flux, and LW↑. (b) Monthly temperature effect due to clouds, estimated from the difference between the measured LW↑ and the estimated clear-sky LW↑ value, for the period January 2011 – October 2013.