# Peer review of "Surface Energy Budget Responses to Radiative Forcing at Summit, Greenland"

_The Cryosphere, 2016_

## Referee Comment (RC1) · P. Kuipers Munneke (Referee) · 3 Nov 2016

Review of "Forcing and responses of the surface energy budget at Summit, Greenland" by Nathaniel Miller et al., submitted for publication in The Cryosphere.

GENERAL

This manuscript presents a multi-year data set of surface energy budget observations, including one year with sufficient observations for a full closure of the budget, from July 2013 to June 2014. While similar estimates and observations have been presented in earlier literature, this paper extends the analysis towards forcings, responses, and the role of clouds and cloud types on the SEB terms and surface temperature.

This paper is clearly written, well illustrated, and a relevant contribution to the recent

surge in literature on the effect of clouds on Greenland climate. The efforts that went into the collection of these rich data definitely warrant publication. However, I feel that the manuscript could benefit from some restructuring, more condensed writing, and some additional analysis. Regarding the latter, I feel a bit of a disconnect between the presentation of the monthly-mean and annual SEB components on one hand (sections 3.3 and further), and the case studies of section 3.2 on the other. It would be worthwhile to improve the connection here, for example by looking at the SEB for different cloud types throughout the season. This illuminates the role of clouds year-round.

Below, I detail my major and minor issues.

MAJOR ISSUES

- Section 3.2 presents a number of observational data sets that are not introduced in the Measurements and Methods section before. This should be added for a proper understanding of the data sets. Specifically, no information is given for the MMCR data, the balloon soundings, cloud radar, and perhaps additional methods that were used in the analysis of cloud cover and type.

- I am somehow struggling with the organization of the results in section 3. The whole section would benefit from some reorganization. In 3.1, surface temperature (being a response to terms in the SEB) is analyzed and discussed. Then, section 3.2 focuses on particular case studies. 3.3 presents annual cycles of SEB fluxes, and 3.4 is about forcings and responses. Personally, I would prefer a structure in which the entire SEB data set is presented first (more or less the current 3.3). After that, the discussion about forcing and responses. And finally, the elucidation of the role of clouds.

- With such a rich data set on cloud properties, it is somewhat disappointing that the analysis in the present manuscript is limited to two - admittedly well chosen - case studies. It would be great if the year-round SEB data set could be split into cloud and non-cloud occurrences and do the analysis on the entire data set. Or bin the results by LWP, by cloud type, etc. This would give even more quantitative insight in the role

of clouds on the SEB throughout the year. It would provide insight in the changes over central Greenland that we may expect in a warming climate.

MINOR ISSUES

P1 L6: what do you mean by "primarily"?

P1 L23: icecap -> ice sheet

P2 L5: there exists newer literature on runoff increase under scenario forcings.

P3 L17: the literature cited here is focused a bit on the work at Utrecht University. There are more observations around the GrIS, like those done at Edinburgh and GEUS in Copenhagen (Denmark).

P3 L23: a more recent example of sublimation analysis from Summit is found in Cullen et al., 2014 (http://onlinelibrary.wiley.com/doi/10.1002/2014JD021557/abstract)

P3 L33: compliment -> complement

P4 L5: remove "in central Greenland"

P4 L11: visa versa -> vice versa

P4 L17: it's -> its

P5 L26: longer term -> longer-term

P6 L13: add Delta to LWP and PWV

P6 L15: hydrometers -> hydrometeors (meteor refers to the Greek word for falling, rather than meter which refers to the Greek word for observing)

P6 L31: I had never heard of the word "thusly" before

P7 L15: Is the linear relation between albedo and Z also used under cloudy conditions? If so, the should be reconsidered as the dependence of albedo on Z vanishes if clouds are sufficiently thick.

[Figure]

P10 L2: Ric -> Ri

P10 L5: very-stable -> very stable

P11 L18: LW derived -> LW-derived

P11 L27: simliar -> similar

P13 L8: boundary-layer -> boundary layer

P13 L32: visa versa -> vice versa

P14 sec 3.3.1: it would be useful here to contrast observations in other studies (from other years and summers) with the numbers you find.

P16 L25: decrease -> decreases

P17 L2: replace the two >'s by <'s.

P20 L25: please provide numbers from the other studies, so that the reader doesn't have to go and look for the differences himself. A table could be useful here.

---

## Referee Comment (RC2) · Anonymous Referee #2 · 21 Nov 2016

The paper combines different ground-based and profile measurement techniques to analyze surface energy fluxes, understand how they are influenced by clouds and their impacts on surface temperature. It provides an important closure of the SEB by calculating turbulent and conductive fluxes and includes a very useful comparison of bulk and EC turbulent flux calculations. To my opinion, the authors have made a tremendous job of combining various high-end measurement techniques to have a closure on SEB through one year of data. At the same time, the authors apply several prior assumptions limiting the learning potential from this rich dataset. The paper should be also condensed and restructured: at times, very lengthy descriptions hide the main idea, while sometimes important information is missing. I recommend this paper for publication given that the major and minor issues below are addressed.

Major comments:

[Figure]

1) It was disappointing to see that the forcing analysis is reduced only to clouds and cloud forcing is reduced only to two cases. A-priory assumptions have been made, eg, indeed liquid-containing clouds are important for SEB, however this is mostly true for summer and ice clouds play an important role in winter (and year total) SEB (as was shown by Van Tricht et al 2016). It would be useful to use these unique comprehensive data to present statistics of SEB depending on a variety of factors - including cloud LWP and IWP (if possible as this parameters is more difficult to derive), PWV, wind speed (especially its effect on turbulent fluxes), near-surface temperature and humidity gradients (and near surface stability).

2) There is no mentioning of the importance of surface snow properties for the surface albedo and its influence on the net SW flux. On p. 7 the authors are saying " The surface albedo is affected by the solar zenith angle" - it is stated that this is the only factor affecting albedo. Have the authors looked at the surface properties? Snowfall, temperature and wind conditions have a large affect on the surface snow microstructure with consequences for the surface albedo (see Carmagnola et al 2013) and thus have to be included into the SEB analysis.

Carmagnola, C. M., Domine, F., Dumont, M., Wright, P., Strellis, B., Bergin, M., Dibb, J., Picard, G., Libois, Q., Arnaud, L., and Morin, S.: Snow spectral albedo at Summit, Greenland: measurements and numerical simulations based on physical and chemical properties of the snowpack, The Cryosphere, 7, 1139-1160, doi:10.5194/tc-7-1139-2013, 2013.

3) It will be useful if the authors extend their linear analysis (fig. 8) to responses to multiple factors. SH and LH strongly depend on the near-surface stability, temperature and humidity gradients, and wind speed. The authors can try multiple regression or neural networks to explore the effect of several predictors.

Minor comments:

Data description has to be made clearer. It will help to include a table with an overview

of all measurements used with their basic characteristics - described in more details in the text.

Some data are described in every detail, while others are just mentioned. For example, the radiosonde data used in the analysis have to be explained including the manufacturer characteristics. There are known biases of humidity measurements at cold temperatures - how do they influence the results?

p. 5: Please describe the instruments at the NOAA/GMF meteorological tower, which, as the authors say, are the primary source of the near-surface measurements

p 5: "The specific humidity at 2 and 10 m, which is needed for deriving LH, is calculated from CIBS relative humidity and temperature measurements in combination with NOAA/GMD temperature and pressure measurements.": what do you mean "in combination"? NOAA and CIBS towers are located 1 km apart... Do you take the average values? How CIBS RH are measured - are these Picarro at 50m tower? What about the data gaps then (they are only until Dec 2013) - are the NOAA RH values used after that? This is not clear.

The description of meteorological measurements has to be made clear. A photograph of both NOAA and CIBS towers will be helpful - as well as a table summarizing all instruments as I mentioned above.

p. 6: " The percent error, using the Picarro measurements as truth, at the 2 and 10 m levels are 53% and 30%, respectively": how were these errors estimated and what are the reasons for such high uncertainty values? are Picarro and meteorological measurements done at the same levels 2 and 10m or as you say the height varies depending on local snow accumulation - and how much is the difference in height then?

You have assumed that Picarro humidity measurements as truth - can you provide more justification? There have been different results of comparing Picarro with independent

humidity measurements and also estimating the accuracy of the field measurements compared to the laboratory measurements (eg Aemisegger et al 2012, Bonne et al 2014). Aemisegger et al 2012 found that the water vapour mixing ratio uncertainty can be quite high in the field and depends on calibration frequency and other effects. I am not an expert in this but invite the authors to include more detailed comments how Picarro measurements were done and used to derive water mixing ratio and their quality. Aemisegger, F., Sturm, P., Graf, P., Sodemann, H., Pfahl, S.,Knohl, A., and Wernli, H.: Measuring variations of 18O and 2H in atmospheric water vapour using two commercial laser-based spectrometers: an instrument characterisation study, Atmos. Meas. Tech., 5, 1491–1511, doi:10.5194/amt-5-1491-2012,2012. Bonne, J.-L., Masson-Delmotte, V., Cattani, O., Delmotte, M.,Risi, C., Sodemann, H., and Steen-Larsen, H. C.: The isotopic composition of water vapour and precipitation in Ivittuut, southern Greenland, Atmos. Chem. Phys., 14, 4419–4439,doi: 10.5194/acp-14-4419-2014, 2014

p. 6: Same comment as for other measurements - please include a table and technical description of the ground-base remote sensing equipment used to derive cloud properties. "in operation since May 2010" - until the present time? no data gaps or measurements issues?

p. 6, section 2.2: where are the ETH radiative sensors are located wrt the NOAA tower and CIBS?

p. 12 section 3.1 title: why mentioning the period in the title? remove it..

p. 12, line 22: "free troposphere above ∼500m": very often boundary layer height (which is the lower value of the free troposphere) in the Arctic extends above 500m and the authors also contradict themselves as on line 17 they speak about synoptic influences at 1-5km

p. 12, section 3.2 title "Case studies" - should reflect more precisely the content (eg, "Cloud forcing case studies")

Section 3.4.2 the text has to be condensed.

Fig. 11: please remove the period from the figure and leave it in the caption

Technical comments:

abstract: .. "calculate estimates of..." - replace with "estimate"

p. 2: trending.. please use a less colloquial word (eg, showing a trend)

2.1 Section title has to be more precise, eg Meteorological and snow measurements

p.5: Root Mean Square -> capitalization not needed

p. 9" which is that determined" - rephrase

p. 12, line 5, last sentence: repetition (rephrase)

p. 13, line 23: on the 10th of November

p. 20, line 8: LWup should be without minus

---

## Author Comment (AC1) · 17 Dec 2016

Referee's comments are in black.

Author's responses are in blue.

Author's changes in the manuscript are in red.

**P. Kuipers Munneke (Referee)**

Review of "Forcing and responses of the surface energy budget at Summit, Greenland" by Nathaniel Miller et al., submitted for publication in The Cryosphere.

GENERAL

This manuscript presents a multi-year data set of surface energy budget observations, including one year with sufficient observations for a full closure of the budget, from July 2013 to June 2014. While similar estimates and observations have been presented in earlier literature, this paper extends the analysis towards forcings, responses, and the role of clouds and cloud types on the SEB terms and surface temperature.

This paper is clearly written, well illustrated, and a relevant contribution to the recent surge in literature on the effect of clouds on Greenland climate. The efforts that went into the collection of these rich data definitely warrant publication. However, I feel that the manuscript could benefit from some restructuring, more condensed writing, and some additional analysis. Regarding the latter, I feel a bit of a disconnect between the presentation of the monthly-mean and annual SEB components on one hand (sections 3.3 and further), and the case studies of section 3.2 on the other. It would be worthwhile to improve the connection here, for example by looking at the SEB for different cloud types throughout the season. This illuminates the role of clouds year-round.

We appreciate the reviewer's detailed comments and suggestions for improvement. Below we explain changes we made and try to clarify the main points of the paper and address your comments and concerns.

Below, I detail my major and minor issues.

MAJOR ISSUES

- Section 3.2 presents a number of observational data sets that are not introduced in the Measurements and Methods section before. This should be added for a proper understanding of the data sets. Specifically, no information is given for the MMCR data, the balloon soundings, cloud radar, and perhaps additional methods that were used in the analysis of cloud cover and type.

The reviewer makes a good point. A lot of the information on the ICECAPS instrumentation and measurements are given in Shupe et. al. 2013 but more information regarding what was done in this paper is needed.

We have included Table 2, which summarizes all the measurements used in this study and added to the text:

"Table 2 summarizes the measurements made by the various instruments described below."

We added more radiosonde information in the measurement section:

"Twice daily Vaisala RS92 radiosondes (0 and 12 UTC) from the Integrated Characterization of Energy, Clouds, Atmospheric State, and Precipitation at Summit (ICECAPS, Shupe et al., 2013b) project are used to directly measure the atmospheric temperature with an uncertainty of $0.5^{\circ}$."

In addition, we moved much of the information from the end of the Measurement section and created a new section "2.6 Cloud Properties and Precipitable Water Vapor".

In section 2.6 we added, "The liquid present cloud fraction for a given month is the number of LWP samples greater than 5 gm^-2 divided by the total number of samples. During May and June 2014 the microwave radiometer measuring 23.84 and 31.40 GHz was off site for repairs and thus LWP and PWV are unavailable for these months. A 35-GHz Millimeter Cloud Radar (MMCR) determines vertically resolved cloud presence. Monthly cloud fractions are calculated using a MMCR detection threshold of -60dBz, retaining sensitivity to most hydrometeors"

We added "MMCR derived cloud fraction (solid) and MWR derived liquid present fraction . . ." to the caption of Figure 6 to link these results to section 2.6.

- I am somehow struggling with the organization of the results in section 3. The whole section would benefit from some reorganization. In 3.1, surface temperature (being a response to terms in the SEB) is analyzed and discussed. Then, section 3.2 focuses on particular case studies. 3.3 presents annual cycles of SEB fluxes, and 3.4 is about forcings and responses. Personally, I would prefer a structure in which the entire SEB data set is presented first (more or less the current 3.3). After that, the discussion about forcing and responses. And finally, the elucidation of the role of clouds.

We think it is important to start with the temperature profiles to show how the surface temperature and subsurface temperatures correlate with the atmospheric temperatures. Then we can delve into how the energy is partitioned in section 3.2 and then investigate factors that affect the variability of the temperature at the surface in section 3.3 and section 3.4.

As you suggest, it is advantageous to cluster the case studies with the forcing and responses analysis. Thus we moved the case studies to after the presentation of the SEB. Yet, we think it is important to put the case studies before the Responses to Surface

Radiative Forcing Section in order to illustrate how the SEB responds to changes in downwelling radation and provide justification for the processes-based relationships.

We reorganized Section 3:

3.1 Temperature

3.2 Surface Energy Budget

3.3 Case Studies

3.4 Responses to Surface Radiative Forcing

To clarify the reasoning behind the new organization of Section 3 we have added a paragraph at the beginning of Section 3.

 "The following observationally based results capture atmospheric/ice sheet interactions. This section will first examine temperature profiles at Summit, providing a foundational understanding for how the atmosphere and snowpack are related. Secondly, investigation of the partitioning of surface energy flux over the annual and diurnal cycles illuminates when various SEB terms are most influential. Finally, quantifying the response of the SEB to changes in downwelling radiation, predominately affected by cloud presence and insolation, shows how the non-radiative SEB terms effect the surface temperature variability. "

We changed the wording between sections to keep continuity of the paper.

The figures were reordered to match changes in Section 3.

- With such a rich data set on cloud properties, it is somewhat disappointing that the analysis in the present manuscript is limited to two - admittedly well chosen - case studies. It would be great if the year-round SEB data set could be split into cloud and non-cloud occurrences and do the analysis on the entire data set. Or bin the results by LWP, by cloud type, etc. This would give even more quantitative insight in the role of clouds on the SEB throughout the year. It would provide insight in the changes over central Greenland that we may expect in a warming climate.

The previous title put too much emphasis on forcing instead of the responses to radiative forcing, the latter being the intended emphasis of this paper. Our approach focuses on how we would expect the SEB to respond to a given change in forcing.  There are a myriad of ways in which clouds can modulate downwelling radiation and we believe it is beyond the scope of this paper to address cloud radiative forcing (CRF) in addition to the responses to the forcing terms. We have linked the current analysis to previous analysis

of CRF in Miller et. al 2015 via the case studies (Fig 5,6), Fig 7 and Fig 11. The process based-relationships go beyond reporting a temporally specific snapshot of the monthly SEB values for a given subset (clear-sky, large LWP, etc . . .). "Process-based relationships distill our understanding of the underlying physical processes into a succinct form that is informative, yet practical. While clouds, the solar cycle, and other processes can influence the downwelling radiation, process relationships between response terms and forcing terms reveal how variability in downwelling radiation affects the other SEB terms. "

Fig 4a shows that clouds are present much of the year (~90% of the time), yet the amount of liquid present has a distinct annual cycle. The magnitude of CRF (fig 11a) is associated with the presence of liquid water. Figure 7a, which bins the forcing terms according LWP and insolation scenarios, supports the conclusion that clouds radiatively warm the surface year round. In this study the forcing terms measured at the surface provide the link to clouds without investigating the myriad of factors that can result in changing the forcing terms at the surface. In fact, Fig 11 gives quantitative insight into the role of clouds on the SEB throughout the year and estimates the resulting effect on the surface temperature.

We have changed the title to "Surface Energy Budget Responses to Radiative Forcing at Summit, Greenland" in order to emphasize the responses of the surface energy budget.

For emphasis we have given the part of the paper that directly ties the new results with the previous CRF results its own subheading titled, "3.4.3 Cloud Effects on the SEB".

MINOR ISSUES

P1 L6: what do you mean by "primarily"

We changed the text to include both methods: "Turbulent sensible heat flux is estimated via the bulk aerodynamic and eddy correlation methods . . ."

P1 L23: icecap -> ice sheet

done

P2 L5: there exists newer literature on runoff increase under scenario forcings.

We added the reference:

Tedesco, M. and Fettweis, X.: 21st century projections of surface mass balance changes for major drainage systems of the Greenland ice sheet, Environ. Res. Lett., 7, 1–11, doi:10.1088/1748-9326/7/4/045405, 2012.

P3 L17: the literature cited here is focused a bit on the work at Utrecht University. There are more observations around the GrIS, like those done at Edinburgh and GEUS in Copenhagen (Denmark).

We added a mass balance reference:

Charalampidis, C., van As, D., Box, J. E., van den Broeke, M. R., Colgan, W. T., Doyle, S. H., Hubbard, A. L., MacFerrin, M., Machguth, H., and Smeets, C. J. P. P.: Changing surface atmosphere energy exchange and refreezing capacity of the lower accumulation area, West Greenland, The Cryosphere, 9, 2163–2181, doi:10.5194/tc-9-2163-2015, 2015.

P3 L23: a more recent example of sublimation analysis from Summit is found in Cullen et al., 2014 (http://onlinelibrary.wiley.com/doi/10.1002/2014JD021557/abstract)

This is a very interesting reference that we overlooked. Thank you for listing it.

We changed the text to include this reference.

"... some studies have targeted SEB annual cycles in 2000-2001 (Cullen, 2003), 2001-2002 (Hoch, 2005), and 2000-2002 (Cullen et al., 2014). "

Cullen, N. J., Mölg, T., Conway, J., and Steffen, K.: Assessing the role of sublimation in the dry snow zone of the Greenland ice sheet in a warming world, J. Geophys. Res. Atmos., 119, 6563–6577, doi:10.1002/2014JD021557, 2014.

P3 L33: compliment -> complement

done

 P4 L5: remove "in central Greenland"

done

P4 L11: visa versa -> vice versa

done

P4 L17: it's -> its

done

P5 L26: longer term -> longer-term

done

P6 L13: add Delta to LWP and PWV

We are already using the Delta symbol in another section of the paper.

We changed the wording to:

" . . . effectively reduce uncertainty in LWP ($\approx 5\ \mathrm{gm}^{-2}$) and PWV ($\approx 0.35$ mm) (Crewell and Löhnert, 2003). "

P6 L15: hydrometers -> hydrometeors (meteor refers to the Greek word for falling, rather than meter which refers to the Greek word for observing)

done

P6 L31: I had never heard of the word "thusly" before

It is a word.
: in this manner :

P7 L15: Is the linear relation between albedo and Z also used under cloudy conditions? If so, the should be reconsidered as the dependence of albedo on Z vanishes if clouds are sufficiently thick.

Yes, you are correct that clouds will increase the albedo compared to clear-sky scene. The parameterization of the upwelling SW is why the uncertainty the upwelling SW went from 1.8% prior to 2014 to 2.8% during 2014. The parameterization was needed to capture the diurnal cycle of SWnet end develop a dataset where all SEB components were estimated on the 30-minute averaged time scale. The relationship between albedo and solar zenith angle was made under all conditions, taking into account all conditions from 2011-2013.

We have reworded this paragraph and added information because it was rather confusing:

"The surface albedo is determined by dividing the measured SW↑ by the measured SW↓ and for clear-sky days should have a minimum at solar noon. During 2014 an asymmetry in the diurnal cycle is observed in the measured albedo, where the albedo in the morning is up to 10% lower than in the evening. The NOAA/GMD measurements, which are mounted on the same fixed arm, indicate the same issue (possibly a gradual slope to the surface due to snow drifts). There is good agreement between the ETH SW↓ measurements and the total direct plus diffuse SW↓ values, suggesting that asymmetry in the diurnal cycle of albedo is likely a problem in the SW↑ component. . . . These uncertainty estimates are larger than the reported uncertainty in the measured SW components of 1.8% (Vuilleumier et al., 2014) because, in addition to Z, albedo is dependent on other factors such as the optical thickness of overlying clouds and surface snow properties."

P10 L2: Ric -> Ri

done

P10 L5: very-stable -> very stable

done

P11 L18: LW derived -> LW-derived

done

P11 L27: simliar -> similar

done

P13 L8: boundary-layer -> boundary layer

done

P13 L32: visa versa -> vice versa

done

P14 sec 3.3.1: it would be useful here to contrast observations in other studies (from other years and summers) with the numbers you find.

Please see the Summary Section and our response to your last comment below.

P16 L25: decrease -> decreases

done

P17 L2: replace the two >'s by <'s.

done

P20 L25: please provide numbers from the other studies, so that the reader doesn't have to go and look for the differences himself. A table could be useful here.

We changed the focus of the SEB data comparison to the Cullen et. al. 2014 manuscript and removed the vague paragraph that was comparing to the earlier Cullen reference studying a similar time period.

"A previous study by Cullen et al. (2014), spanning the time period 17 June 2000 – 18 June 2002, also reports the annual cycle of the surface energy budget components at Summit Station. Comparing the annual mean values of this study to the earlier study reveals that Q is 6.8 $\text{Wm}^{-2}$ smaller and SH, LH and G are $1.6, 0.9$, and $4.8 \text{ Wm}^{-2}$ larger, respectively. The differences in the annual mean values could be due to possible decreases in cloud cover (Comiso and Hall, 2014), since the recent annual forcing value

is 7.3 Wm$^{-2}$ smaller than the 190.1 Wm$^{-2}$ reported by Cullen et al. (2014). July 2014 had the largest occurrence of liquid-bearing clouds for the current study resulting in an average Q of 6.1 Wm$^{-2}$ compared to 15.6 Wm$^{-2}$ reported by Cullen et al. (2014). The July 2014 forcing terms are 265.3 Wm$^{-2}$ compared to 268.0 Wm$^{-2}$ in 2000 - 2002, suggesting that a 6.8 Wm$^{-2}$ increase in LW↑ is likely due to synoptically-driven warmer air masses above Summit Station in 2014 and not due to changes in cloud radiative forcing."

We added Section 5, making the current dataset easily available, allowing the reader to compare our results with other studies.

"The surface energy budget dataset is available online in the National Science Foundation's Arctic Data Center. [Matthew Shupe and Nathaniel Miller. 2016. Surface energy budget at Summit, Greenland. NSF Arctic Data Center. doi:10.18739/A2Z37J]"

---

## Author Comment (AC2) · 17 Dec 2016

Referee's comments are in black.

Author's responses are in blue.

Author's changes in the manuscript are in red.

**Anonymous Referee #2**

The paper combines different ground-based and profile measurement techniques to analyze surface energy fluxes, understand how they are influenced by clouds and their impacts on surface temperature. It provides an important closure of the SEB by calculating turbulent and conductive fluxes and includes a very useful comparison of bulk and EC turbulent flux calculations. To my opinion, the authors have made a tremendous job of combining various high-end measurement techniques to have a closure on SEB through one year of data. At the same time, the authors apply several prior assumptions limiting the learning potential from this rich dataset. The paper should be also condensed and restructured: at times, very lengthy descriptions hide the main idea, while sometimes important information is missing. I recommend this paper for publication given that the major and minor issues below are addressed.

Thank you for the detailed reading of the paper, valuable comments and suggestions. We have tried to clarify points of confusion and add context when needed in order to make the paper stronger.

Major comments:

1) It was disappointing to see that the forcing analysis is reduced only to clouds and cloud forcing is reduced only to two cases. A-priory assumptions have been made, eg, indeed liquid-containing clouds are important for SEB, however this is mostly true for summer and ice clouds play an important role in winter (and year total) SEB (as was shown by Van Tricht et al 2016). It would be useful to use these unique comprehensive data to present statistics of SEB depending on a variety of factors - including cloud LWP and IWP (if possible as this parameters is more difficult to derive), PWV, wind speed (especially its effect on turbulent fluxes), near-surface temperature and humidity gradients (and near surface stability).

The previous title prompted expectations that the paper would specifically address forcing at Summit. The intended focus of the paper is rather the responses to radiative forcing and how we would expect the SEB to respond to a given change in forcing. There are a number of ways in which clouds can modulate the downwelling radiation at the surface and we believe it is beyond the scope of this paper to address both radiative forcing and the responses. The case studies (Fig 5,6), Fig 7 and Fig 11 link the current analysis to previous analysis of cloud radiative forcing in Miller et. al 2015. We believe the process based-relationships

are useful for model validation. The relationships go beyond reporting a temporally specific snapshot of the monthly SEB values for a given subset (clear-sky, large LWP, etc . . . ). Often models do not accurately capture cloud occurrence and phase. This observationally based data set can be used to validate the energy fluxes at the surface and go beyond simply looking at individual components such as the downwelling LW and investigate if the non-radiative terms would respond realistically to produce accurate surface temperatures.

The title has been changed to: Surface Energy Budget Responses to Radiative Forcing at Summit, Greenland.

It is true that ice clouds play an important role in determining the SEB. Fig 4a shows that clouds are present much of the year (~90% of the time), while the amount of liquid present has a distinct annual cycle. Thus it is true that ice clouds are important in radiatively warming the surface yet cloud phase is the dominant factor in determining the magnitude of the warming. Both ice clouds and mixed-phase clouds modulate SW radiation and thus solar zenith angle is an important consideration in CRF. Figure 7, which bins the forcing terms according LWP and insolation scenarios, supports the conclusion that clouds radiatively warm the surface year round, during times of high solar elevation and in the absence of insolation.

For emphasis we created a new subheading titled, "3.4.3 Cloud Effects on the SEB"
This subsection provides estimates of quantitative insights (Fig11) into the role of clouds on the SEB throughout the year and estimates the resulting effect on surface temperature.

2) There is no mentioning of the importance of surface snow properties for the surface albedo and its influence on the net SW flux. On p. 7 the authors are saying " The surface albedo is affected by the solar zenith angle" - it is stated that this is the only factor affecting albedo. Have the authors looked at the surface properties? Snowfall, temperature and wind conditions have a large affect on the surface snow microstructure with consequences for the surface albedo (see Carmagnola et al 2013) and thus have to be included into the SEB analysis.

Carmagnola, C. M., Domine, F., Dumont, M., Wright, P., Strellis, B., Bergin, M., Dibb, J., Picard, G., Libois, Q., Arnaud, L., and Morin, S.: Snow spectral albedo at Summit, Greenland: measurements and numerical simulations based on physical and chemical properties of the snowpack, The Cryosphere, 7, 1139-1160, doi:10.5194/tc-7-1139- 2013, 2013.

The reviewer brings up a good point that the albedo is affected by snow surface properties. In our analysis measured changes in albedo affect the SWnet component of the forcing term. It is estimated that the changes to the SWdownwelling and LWdownwelling by clouds and insolation are much greater than the changes to SWupwelling by albedo changes alone in the dry snow zone so we focus on clouds and insolation. Yet, the variability of the albedo for a given month will also affect the forcing terms and can be considered a forcing which induces a response of the other SEB terms.

We added this to Section 3.4 to reflect this sentiment:

"In addition, variability in surface albedo acts as a forcing, although at Summit the magnitude of downwelling radiation variations are much greater than the effect of albdeo variations on forcing terms. "

Ideally we would rely on the measurements to determine SWnet. Yet, a parameterization was needed to capture the diurnal cycle of SWnet and develop a dataset where all SEB components were estimated on the 30-minute averaged time scale.  The relationship between albedo and solar zenith angle was made under all conditions, taking into account all conditions from 2011-2013.  The parameterization of the upwelling SW is why the uncertainty the upwelling SW went from 1.8% prior to 2014 to 2.8% during 2014.

We have reworded this paragraph and added information because it was rather confusing.

"The surface albedo is determined by dividing the measured SW↑ by the measured SW↓ and for clear-sky days should have a minimum at solar noon. During 2014 an asymmetry in the diurnal cycle is observed in the measured albedo, where the albedo in the morning is up to 10% lower than in the evening. The NOAA/GMD measurements, which are mounted on the same fixed arm, indicate the same issue (possibly a gradual slope to the surface due to snow drifts). There is good agreement between the ETH SW↓ measurements and the total direct plus diffuse SW↓ values, suggesting that asymmetry in the diurnal cycle of albedo is likely a problem in the SW↑ component.  . . . These uncertainty estimates are larger than the reported uncertainty in the measured SW components of 1.8% (Vuilleumier et al., 2014) because, in addition to Z, albedo is dependent on other factors such as the optical thickness of overlying clouds and surface snow properties."

3) It will be useful if the authors extend their linear analysis (fig. 8) to responses to multiple factors. SH and LH strongly depend on the near-surface stability, temperature and humidity gradients, and wind speed. The authors can try multiple regression or neural networks to explore the effect of several predictors.

We believe multiple regression or neural networks analysis is beyond the scope of the paper as we intend to focus on the SEB responses to the forcing terms, while building off previous research that estimates the surface energy fluxes from these predictors. In fact, the effect of these predictors is folded into the values of the turbulent energy fluxes because the input includes stability corrections, wind speed, and temperature and humidity (LH only) gradients.

It is true that our linear analysis is looking at only the first order influence, which is radiative. The RMSE values (Fig 10) of the linear fit are due to the higher order effects (such as mechanical mixing due to high winds or decreases in turbulence due to high stability). In order to incorporate the uncertainties associated with both a response term and the forcing terms we have redone the linear analysis using a different linear fit routine. This technique provides a more accurate response and a smaller uncertainty in the slope by accounting for measurement uncertainties in both x and y.

Table 1 was updated with baseline uncertainties that were not included in the previous version and text was added to describe the linear regression used.

"Performing a linear fit (fitexy, Press et al., 1992) on the relationship between the forcing and response terms, which includes uncertainties in both terms, yields a slope of -1.01 (Figure 8a), indicating that the SEB is largely radiatively driven, . . ."

Press, W. H., Teukolsky, S. A., Vetterling, W. T., and Flannery, B. P.: Numerical Recipes in C: The Art of Scientific Computing, University Press, Cambridge, 2nd edn., 1992.

We recalculated the linear fits using improved linear fit routine, updated Figures 8-11, and made appropriate text changes to ensure consistency across the full document.

Minor comments: Data description has to be made clearer. It will help to include a table with an overview of all measurements used with their basic characteristics - described in more details in the text.

Some data are described in every detail, while others are just mentioned. For example, the radiosonde data used in the analysis have to be explained including the manufacturer characteristics. There are known biases of humidity measurements at cold temperatures - how do they influence the results?

We acknowledge that a clear description of the data is important and have attempted to make the appropriate changes to the document to improve the data description in the manuscript.

We added Table 2, which summarizes all the measurements used in this study.

"Table 2 summarizes the measurements made by the various instruments described below."

Possible biases in the radiosonde humidity measurements do not affect our analysis because the humidity profile is not used in this study.

We added more radiosonde information in the measurement section and removed the mention of humidity measurements:

"Twice daily Vaisala RS92 radiosondes (0 and 12 UTC) from the Integrated

Characterization of Energy, Clouds, Atmospheric State, and Precipitation at Summit (ICECAPS, Shupe et al., 2013b) project are used to directly measure the atmospheric temperature with an uncertainty of 0.5°.”

In addition, we moved much of the information from the end of the Measurement section and created a new section “2.6 Cloud Properties and Precipitable Water Vapor”.

To address our shortcoming of inadequately describing the cloud property measurements, in section 2.6 we added: “The liquid present cloud fraction for a given month is the number of LWP samples greater than 5 gm^-2 divided by the total number of samples. During May and June 2014 the microwave radiometer measuring 23.84 and 31.40 GHz was off site for repairs and thus LWP and PWV are unavailable for these months. A 35-GHz Millimeter Cloud Radar (MMCR) determines vertically resolved cloud presence. Monthly cloud fractions are calculated using a MMCR detection threshold of -60dBz, retaining sensitivity to most hydrometeors”

 “MMCR derived cloud fraction (solid) and MWR derived liquid present fraction . . .” was added to the caption of Figure 6 to link these results to section 2.6.

p. 5: Please describe the instruments at the NOAA/GMF meteorological tower, which, as the authors say, are the primary source of the near-surface measurements

We added the sensor information in Table 2 and added in the text:

“. . . temperature measurements (Logan RTD - PT139 special order) with a specified resolution of . . .”

p 5: "The specific humidity at 2 and 10 m, which is needed for deriving LH, is calculated from CIBS relative humidity and temperature measurements in combination with NOAA/GMD temperature and pressure measurements.": what do you mean "in combination"? NOAA and CIBS towers are located 1 km apart... Do you take the average values? How CIBS RH are measured - are these Picarro at 50m tower? What about the data gaps then (they are only until Dec 2013) - are the NOAA RH values used after that? This is not clear.

The description of meteorological measurements has to be made clear. A photograph of both NOAA and CIBS towers will be helpful - as well as a table summarizing all instruments as I mentioned above.

The wording you reference in the manuscript was confusing regarding how specific humidity was determined. The 30-minute averaged temperature and RH values were used to calculate the vapor pressure via calculating the saturation vapor pressure using the Goff-Gratch formulation.

 “ The specific humidity at 2 and 10 m, which is needed for deriving LH, is calculated from the CIBS relative humidity, CIBS temperature and NOAA/GMD pressure measurements. The saturation vapor pressure, at a given temperature, is calculated using

the Goff-Gratch formulation and then multiplied by the relative humidity to get the vapor pressure. Specific humidity is proportional to the ratio of the vapor pressure to the difference in vapor pressure and air pressure. To provide continuity in the LH estimates the meteorologically derived specific humidity values are used as input to the LH flux calculations, while direct measurements of water vapor are used to estimate the uncertainty in this technique during overlapping time periods. . ."

p. 6: " The percent error, using the Picarro measurements as truth, at the 2 and 10 m levels are 53% and 30%, respectively": how were these errors estimated and what are the reasons for such high uncertainty values? are Picarro and meteorological measurements done at the same levels 2 and 10m or as you say the height varies depending on local snow accumulation - and how much is the difference in height then?

You have assumed that Picarro humidity measurements as truth - can you provide more justification? There have been different results of comparing Picarro with independent humidity measurements and also estimating the accuracy of the field measurements compared to the laboratory measurements (eg Aemisegger et al 2012, Bonne et al 2014). Aemisegger et al 2012 found that the water vapour mixing ratio uncertainty can be quite high in the field and depends on calibration frequency and other effects. I am not an expert in this but invite the authors to include more detailed comments how Picarro measurements were done and used to derive water mixing ratio and their quality. Aemisegger, F., Sturm, P., Graf, P., Sodemann, H., Pfahl, S.,Knohl, A., and Wernli, H.: Measuring variations of 18O and 2H in atmospheric water vapour using two commercial laser-based spectrometers: an instrument characterisation study, At- mos. Meas. Tech., 5, 1491–1511, doi:10.5194/amt-5-1491-2012,2012. Bonne, J.-L., Masson-Delmotte, V., Cattani, O., Delmotte, M.,Risi, C., Sodemann, H., and Steen- Larsen, H. C.: The isotopic composition of water vapour and precipitation in Ivittuut, southern Greenland, Atmos. Chem. Phys., 14, 4419–4439,doi: 10.5194/acp-14-4419- 2014, 2014

We have calculated percent error, thusly:

Percent error = 100*(q_met – q_picarro)/q_picarro

Meteorologically derived percent errors are large because the uncertainties in q_met are large relative to the amount of water vapor that exists in such a dry environment. The Picarro, CIBS temperature and CIBS RH are all mounted on the 50m tower so any change in local accumulation will uniformly affect them all.

The reviewer is correct in saying the Picarro measurements are not absolute truth. Yet, they are well calibrated and thus we compute the percent error of the meteorologically derived values using the calibrated values. The details of the calibration are beyond the scope of this paper so we have provided the Bailey et. al. 2015 reference.

We changed the text reflect the fact that the percent error is calculated in reference to the Picarro measurements and removed the word "truth".

"The percent error, relative to the Picarro measurements, at the 2 and 10 m levels . . ."

p. 6: Same comment as for other measurements - please include a table and technical description of the ground-base remote sensing equipment used to derive cloud properties. "in operation since May 2010" - until the present time? no data gaps or measurements issues?

We have added information in Section 2.6 and summarized the instruments in Table 2.

p. 6, section 2.2: where are the ETH radiative sensors are located wrt the NOAA tower and CIBS?

All measurements are with in 1km of each other, thus 30-minute averages should account for much of the local variability on this relatively homogeneous surface.

The relative position of the ETH measurements is described in the text: "The radiation station is located between the 50m tower and the NOAA/GMD met tower."

p. 12 section 3.1 title: why mentioning the period in the title? remove it.

Changed title to: "Temperature Profiles"

p. 12, line 22: "free troposphere above ~500m": very often boundary layer height (which is the lower value of the free troposphere) in the Arctic extends above 500m and the authors also contradict themselves as on line 17 they speak about synoptic influences at 1-5km

Changed to: "free troposphere above ~1 km"

p. 12, section 3.2 title "Case studies" - should reflect more precisely the content (eg, "Cloud forcing case studies")

done.

Section 3.4.2 the text has to be condensed.

These are the main findings of the paper and thus we left much of this analysis intact.

A new subsection (3.4.3) was created to separate the estimated cloud effects from the SEB response analysis.

Fig. 11: please remove the period from the figure and leave it in the caption

done

Technical comments:

abstract: .. "calculate estimates of..." - replace with "estimate"

done

p. 2: trending.. please use a less colloquial word (eg, showing a trend)

done

2.1 Section title has to be more precise, eg Meteorological and snow measurements

done

p.5: Root Mean Square -> capitalization not needed

done

p. 9" which is that determined" – rephrase

changed to: "which was determined"

p. 12, line 5, last sentence: repetition (rephrase)

We combined the sentences to remove the repetitive nature of the sentences. "A warm or cold pulse at the surface propagates to deeper portions of the GIS over time and can take days to influence the temperatures at 1-2 m depth."

p. 13, line 23: on the 10th of November

done

p. 20, line 8: LWup should be without minus

done